# Guided and Interpretable Neural Operator Design for Partial Differential Equation Learning

## Abstract

Accurate numerical solutions of partial differential equations (PDEs) are crucial in numerous science and engineering applications. In this work, we introduce a novel neural PDE solver named AFDONet, which incorporates neural operator learning and adaptive Fourier decomposition (AFD) theory for the first time into a specifically designed variational autoencoder (VAE) structure, to solve a general class of nonlinear PDEs on smooth manifolds. AFDONet is the first neural PDE solver whose architectural and component design is fully guided by an established mathematical framework (in this case, AFD theory), turning neural operator design from an art to a science. Thus, AFDONet also exhibits exceptional mathematical explainability and groundness, and enjoys several desired properties. Furthermore, AFDONet achieves outstanding solution accuracy and competitive computational efficiency in several benchmark problems. In particular, thanks to its deep connections with AFD theory, AFDONet shows superior performance in solving PDEs on i) arbitrary (Riemannian) manifolds, and ii) datasets with sharp gradients. Overall, this work presents a new paradigm for designing explainable neural operator frameworks.

## 1 Introduction

A wide range of scientific and engineering phenomena can be characterized and modeled by partial differential equations (PDEs). Most nonlinear PDEs do not have analytical solutions and need to be solved numerically. Traditional discretization-based numerical solvers, such as finite element methods (FEM) and finite difference methods (FDM), can become quite slow, inefficient, and unstable (Hittinger & Banks, 2013; Sokic et al., 2011; Carey et al., 1993). On the other hand, data-driven methods, such as neural PDE solvers, can directly learn the trajectory of the family of equations from the data, and thus can be orders of magnitude faster than traditional solvers Li et al. (2020). Most neural PDE solvers operate either by approximating the solutions (Raissi et al., 2019; Han et al., 2018), directly learning the mappings between function spaces (Li et al., 2020; Li & Ye, 2025; Tripura & Chakraborty, 2023; Lu et al., 2020), or integrating neural networks with conventional numerical solvers in a hybrid manner (Bar-Sinai et al., 2019; Li et al., 2025; Brevis et al., 2020).

While most existing PDE solvers are designed for regular Euclidean domains, in many real-world applications, PDEs are defined on non-Euclidean manifolds. Most existing approaches to solve PDEs on manifolds rely on classical numerical approaches, such as parameterization (Lui et al., 2005), collocation (Chen & Ling, 2020), and spectral methods (Yan et al., 2023). Although researchers have begun to explore manifold-aware neural architectures that can learn directly from point clouds (He et al., 2024; Liang et al., 2024) or graphs (Bronstein et al., 2017), they cannot easily be generalized to different manifolds. Thus, extending neural PDE solvers to manifold domains remains challenging. Instead, pullback operators are often used in existing neural PDE solvers to map functions and differential operators from the manifold to a Euclidean space.

Another research gap in neural operator solver is that, so far, the design of exact neural architectures in many neural PDE solvers has been "more of an art than a science" (Sanderse et al., 2025). Typically, neural architecture design is done in a bottom-up approach that involves significant intu-

ition, expert experience, and trial-and-error experimentation. And rigorous mathematical basis and explainability have been lacking in guiding the design of these neural architectures.

**Our approach.** To bridge these gaps, in this work, we propose a novel neural PDE solver named AFDONet for solving general nonlinear PDEs on smooth manifolds. Specifically, AFDONet is a variational autoencoder (VAE)-based neural operator whose design replicates adaptive Fourier decomposition (AFD), a novel signal decomposition technique achieving higher accuracy and significant computational speedup compared to conventional signal decomposition methods (Qian, 2010). AFD can approximate signals and functions in a reproducing kernel Hilbert space (RKHS) on different domains and manifolds (Qian et al., 2011; 2012; Zhang et al., 2023; Song & Sun, 2022), making it a desirable choice for designing theory-guided, interpretable neural operator for solving PDEs on manifolds. Motivated by this, in AFDONet, latent variables are first mapped to their nearest reproducing kernel Hilbert space (RKHS) via a latent-to-RKHS network, followed by reconstructing the solution manifold using a new type of decoder replicating AFD operations.

**Key contributions.** The key contributions of this work are summarized as follows:

1. We follow a unique, top-down approach based on adaptive Fourier decomposition (AFD) theory to guide every step in the design of AFDONet's neural architecture. This presents a new paradigm for designing explainable neural operator frameworks.

2. AFDONet is mathematically grounded in AFD theory, as the solutions produced by our novel neural architecture can be interpreted as an adaptive decomposition into basis functions. Thus, AFDONet has rigorous mathematical foundations based on approximation theory and possesses several desirable properties.

3. We demonstrate the effectiveness of our AFDONet solver by comparing its solution accuracy with several neural PDE solvers over benchmark problems on arbitrary (Riemannian) manifolds and datasets with sharp gradients. We show that AFDONet achieves outstanding performance in terms of solution accuracy and its capability to reconstruct solution manifolds.

## 2 PROBLEM STATEMENT

We consider a PDE defined on a spatial domain $\Omega \subset \mathbb{R}^d$ and a time interval $(0, T]$:

$$\mathcal{L}_\alpha[u(x,t)] = f(x,t), \quad \forall (x,t) \in \Omega \times (0, T], \tag{1}$$

where $\mathcal{L}$ denotes the differential operator, $f(x,t)$ is the source/sink term, and the parameter function $\alpha \in \mathcal{A}$ specifies the physical parameters and the initial and boundary conditions. Our goal is to learn a neural operator $G : \mathcal{A} \to \mathcal{F}(D \times [0, T])$, which maps the parameter function $\alpha$ from its parameter space $\mathcal{A}$ to the corresponding solution $u(x,t) \in \mathcal{F}$. In this work, we focus on two types of tasks: (i) the static task, which solves a PDE for one set of physical parameters $\alpha$ and a fixed final time $T$ (i.e., $u(x, T)$); and (ii) the autoregressive task, which forecasts the PDE solution at time step $t + 1$ (i.e., $u(x, t + 1)$) based on the solution at the previous time step $t$ (i.e., $u(x, t)$).

## 3 RELATED WORK

**Classic Fourier-based methods**, such as Fourier transform approaches (Negero, 2014), Fourier series expansions (Asmar, 2016), and Fourier spectral methods (Alali & Albin, 2020), have been extensively used to solve PDEs numerically. Classic Fourier-based methods offer accurate and efficient representations of smooth, periodic functions by transforming differential operators into simple algebraic operations in the frequency domain. However, the use of global basis functions produces oscillations when approximating functions with discontinuities or sharp transitions (Gottlieb & Shu, 1997). Furthermore, the fixed basis structure in these methods lacks adaptability to signals with time-localized, transient, or nonperiodic features. In addition, these methods are typically defined on simple, regular domains, making them difficult to apply directly to manifolds.

**Operator learning** aims to directly learn the mapping between infinite-dimensional function spaces (e.g., from input functions to solutions) to enable fast, mesh-independent approximation of PDE solutions across various input conditions, including source and/or sink term, physical parameters,

and initial and boundary conditions. Among existing operator learning-based PDE solvers, two notable ones backed by the approximation theory are DeepONet (Lu et al., 2019; 2021), which is inspired by the universal approximation theorem for nonlinear operators, and the Fourier Neural Operator (FNO) (Li et al., 2020; 2023b), which performs convolution in the frequency domain to capture global spatial dependencies efficiently. Both operator learning paradigms have led to several new variants. Some of the recently developed network architectures (He et al., 2023; Goswami et al., 2022; He et al., 2024; Li et al., 2023a) built upon DeepONet provide enhancements such as physics-informed structure, parameterized geometry and phase-field modeling. Some of the new variants of FNO include Factorized FNO (F-FNO) (Tran et al., 2021), Decomposed FNO (D-FNO) (Li & Ye, 2025), Spherical FNO (Bonev et al., 2023), Domain Agnostic FNO (DAFNO) (Liu et al., 2023), Wavelet Neural Operator (WNO) (Tripura & Chakraborty, 2023), Multiwavelet Neural Operator (MWT) (Gupta et al., 2021), Coupled Multiwavelet Neural Operator (CMWNO) (Xiao et al., 2025), and Adaptive Fourier Neural Operator (AFNO) (Guibas et al., 2021).

**Physics-informed representation learning and variational autoencoder (VAE).** Another avenue for solving PDEs is to directly incorporate physical knowledge and constraints derived from the PDE into a neural architecture. One of the popular frameworks is the Physics-Informed Neural Network (PINN) (Raissi et al., 2019; 2017), where the PDE itself is embedded in the loss function as a regularization term. Another approach is to introduce variational autoencoders (VAEs) (Tait & Damoulas, 2020; Kingma et al., 2013) in a physics-informed architecture. This provides a structured latent space and a probabilistic framework for integrating physics, leading to more stable and generalizable representation learning. Several physics-informed VAE models have recently been proposed, including Glyn-Davies et al. (2024); Zhong & Meidani (2023); Takeishi & Kalousis (2021); Lu et al. (2020). Specifically, Lu et al. (2020) used a dynamics encoder and a propagating decoder to extract interpretable physical parameters from PDEs. Later, Takeishi & Kalousis (2021) proposed a physics-informed VAE model by introducing physics-based models to augment latent variables, encoder, and decoder. However, these methods lack rigorous theoretical justifications for the design of their neural architectures that ensure convergence and performance guarantees.

## 4 PRELIMINARIES TO ADAPTIVE FOURIER DECOMPOSITION (AFD)

AFD is a novel signal decomposition technique that leverages the Takenaka-Malmquist system and adaptive orthogonal bases (Qian, 2010; Qian et al., 2012). It is established as a new approximation theorem in a reproducing kernel Hilbert space (RKHS) sparsely in a given domain $\Omega$ as $s = \sum_{i=1}^{\infty} \langle s, \mathscr{B}_i \rangle \mathscr{B}_i$ for the chosen orthonormal bases $\mathscr{B}_i$ (Saitoh et al., 2016). An RKHS is a Hilbert space of functions where evaluation at any point is continuous with respect to the inner product $\langle \cdot, \cdot \rangle$, and each point on the domain corresponds to a unique kernel function. For AFD in RKHS, the sparse bases $\{\mathscr{B}_i\}_i$ are made orthonormal to each other by applying Gram-Schmidt orthogonalization to the normalized reproducing kernels associated with a set of adaptively selected "poles" $\{a_i\}_i$, which are complex numbers used to parameterize the sparse bases. Specifically, to decompose signals in a Hardy space (i.e., a Hilbert space consisting of holomorphic functions defined on the unit disk), which can be further relaxed to an RKHS (Song & Sun, 2022), the orthonormal basis functions $\mathscr{B}_i$ can be derived as:

$$\mathscr{B}_i(z) = \frac{\sqrt{1 - |a_i|^2}}{1 - \overline{a_i} z} \prod_{j=1}^{i-1} \frac{z - a_j}{1 - \overline{a_j} z}, \quad a_i \in \mathbb{D}, \tag{2}$$

where $\mathbb{D} = \{z \in \mathbb{C} : |z| < 1\}$. To adaptively select the sequence of poles such that convergence of AFD approximation is ensured, one shall follow the so-called "maximal selection principle", such that the resulting $|\langle s, \mathscr{B}_i \rangle|$ is as large as possible. That is, to select the next pole $a_i$ given $i - 1$ already selected poles, $a_1, \ldots, a_{i-1}$ (hence bases $\mathscr{B}_1, \ldots, \mathscr{B}_{i-1}$), the corresponding orthonormal basis $\mathscr{B}_i$ needs to satisfy:

$$|\langle s, \mathscr{B}_i \rangle| \geq \rho_i \sup \left\{ \langle s, \mathscr{B}_i' \rangle | b_i \in \Omega \backslash \{a_1, \ldots, a_{i-1}\} \right\}, \tag{3}$$

where $0 < \rho_0 \leq \rho_i < 1$, $\mathscr{B}_1' = \frac{k_{b_1}}{\|k_{b_1}\|_{H(\Omega)}}$ and $\mathscr{B}_i' = \frac{k_{b_i} - \sum_{j=1}^{i-1} \langle k_{b_i}, \mathscr{B}_j \rangle \mathscr{B}_j}{\|k_{b_i} - \sum_{j=1}^{i-1} \langle k_{b_i}, \mathscr{B}_j \rangle \mathscr{B}_j\|_{H(\Omega)}}$. Here, $k_{b_i}$ is the reproducing kernel (e.g., Gaussian or Bergman kernel) at $b_i$. In classic AFD theory, the algorithmic procedure of pole selection, which is discussed in Song & Sun (2022), is computationally expensive.

Therefore, integrating the classical AFD with neural operators is a promising approach to enable fast and accurate solution of PDEs through the use of adaptive orthonormal basis functions.

## 5 AFDONet Architecture

Guided by the AFD theory, we design AFDONet to approximate PDE solution spaces on any smooth manifold. The AFDONet architecture shown in Figure 1 consists of an encoder, a latent-to-RKHS network, and an AFD-type dynamic convolutional kernel network (CKN). These components work synergistically to enhance the performance of the AFDONet solver. After the encoder, AFDONet identifies the closest RKHS where the latent variables reside using a latent-to-RKHS network. Subsequently, AFDONet reconstructs the PDE solutions by replicating the AFD operation and adaptively selecting the poles using a specially designed decoder network. For static tasks, the training dataset is denoted as $\{u(x, T)\}_{\{\alpha\}}$ for different sets of physical parameters $\alpha$, while for autoregressive tasks, the training dataset is denoted as $\{u(x, t), u(x, t + 1)\}_{t=0}^T$.

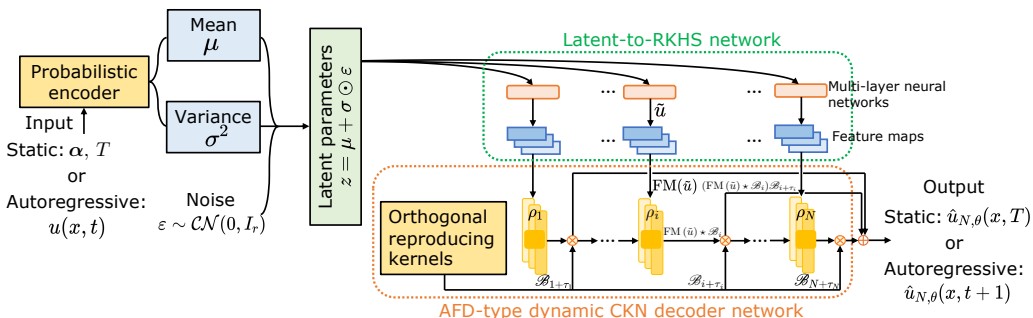

Figure 1: Our proposed AFDONet framework, which adopts VAE as the backbone, introduces a latent-to-RKHS network and a dynamic CKN decoder to reproduce the AFD setting and operation.

**The use of VAE as architecture backbone** is motivated from both methodological and experimental perspectives. From a methodological perspective, the use of VAE architecture as the backbone for our AFDONet is motivated by several reasons. First, many PDE solution fields lie on low-dimensional manifolds in high-dimensional function space. VAE-based neural operators can learn a probabilistic latent representation of these manifolds, mapping high-dimensional inputs to a compact latent space while capturing variation in solution behavior. This reduces the complexity of learning and enables generalization across parametric inputs, as shown in many prior successes in VAE-based neural operators (Zhong & Meidani, 2023; Rafiq et al., 2025; Lu et al., 2020; Takeishi & Kalousis, 2021). Second, VAE is inherently connected to AFD theory in several ways. For instance, VAEs benefit from frequency transformations (Li et al., 2024), which are the foundation of bases used in AFD. Also, the maximal selection principle of basis functions in AFD aligns well with the variational inference of VAE (Chen et al., 2020a).

From an experimental perspective, we will show in Section 7 that the use of VAE and its holistic integration with other components in the AFDONet architecture help significantly improve the accuracy of PDE solutions on manifolds.

**The encoder network** maps the inputs $\alpha$ or $u(x, t)$ to a latent space in the complex domain $\mathbb{C}^{2r}$ using a standard probabilistic encoder network based on the VAE framework. For the static task, this means:

$$\left(\mu(\alpha), \log \sigma^2(\alpha)\right) = A_2\big(\Phi(A_1 \alpha)\big), \quad z = \mu(\alpha) + \sigma(\alpha) \odot \varepsilon, \quad \varepsilon \sim \mathcal{CN}(0, I_r), \quad (4)$$

where $A_1 \in \mathbb{C}^{W_e \times d}$ and $A_2 \in \mathbb{C}^{2r \times W_e}$ are the weight matrices (where $W_e = \mathcal{O}(r)$), $\Phi(\cdot)$ is the activation function, the latent mean is $\mu(\alpha) \in \mathbb{C}^r$, the log-variance is $\log \sigma^2(\alpha) \in \mathbb{C}^r$, and $z$ is the latent parameter vector.

For the autoregressive task, the input $u_t = u(x, t)$ lies on the Hilbert space $H(\mathcal{M})$ of manifold $\mathcal{M}$. Therefore, $u_t = u(x, t)$ must be projected from $H(\mathcal{M})$ into an appropriate complex domain. Let

$\{\phi_k\}_{k=0}^{\infty}$ be an orthonormal Fourier basis. Then, we define a linear projection:

$$\Pi_K u_t := \big(\langle u_t, \phi_0 \rangle, \ldots, \langle u_t, \phi_{K-1} \rangle\big) \in \mathbb{C}^K, \tag{5}$$

which retains the first $K$ modes of the field. This leads to the following encoder structure:

$$\big(\mu_t, \log \sigma_t^2\big) = A_2\Big(\Phi\big(A_1 \Pi_K u_t\big)\Big), \quad z_t = \mu_t + \sigma_t \odot \varepsilon_t, \quad \varepsilon_t \sim \mathcal{CN}(0, I_r), \tag{6}$$

where $A_1 \in \mathbb{C}^{W_e \times K}$ and $A_2 \in \mathbb{C}^{2r \times W_e}$ are the weight matrices (where $W_e = \mathcal{O}(r)$), $\Phi(\cdot)$ is the activation function. In both tasks, the encoder network has a depth $L_e = 2$ and width $W_e = \mathcal{O}(r)$.

**The latent-to-RKHS network** maps the latent parameters to convolutional kernels while constraining the corresponding functional space to be an RKHS, where the AFD operations are defined. This extends the latent-to-kernel network proposed by Lu et al. (2020) by explicitly accounting for the fact that the kernels are constructed in a Hilbert space. Our latent-to-RKHS network consists of multi-layer fully-connected feedforward (MLP) networks and feature maps. The MLP networks will first take the latent parameter vector $z$ obtained from the encoder network to generate $\tilde{u}(x, \cdot)$ on $H(\mathcal{M})$. Then, feature maps $\mathrm{FM}(\cdot)$ will map $\tilde{u}(x, \cdot)$ to its nearest RKHS $\mathcal{H}(\mathcal{M})$ via orthogonal projection. This way, the latent-to-RKHS network learns the feature maps from $H(\mathcal{M})$ to its nearest RKHS $\mathcal{H}(\mathcal{M})$, in which the reproducing kernel $k_a$ can be obtained by:

$$k_a(\xi) = \sum_{i=1}^{N'} \nu_i(a) \, e^{2\pi j \phi \cdot (\xi - y_i)}, \qquad \forall a, \xi \in \mathcal{M} \tag{7}$$

where $j^2 = -1$ and $\phi$ is the fundamental frequency. Here, weights $\nu_i \in \mathbb{C}$ and parameters $y_i \in \mathcal{M}$ are learnable from the latent-to-RKHS network. Essentially, a feature map applies a fast Fourier transform (FFT) to its input, multiplies the top $N'$ low-frequency components by learnable complex weights while discarding the high-frequency components, and then performs an inverse FFT. Note that this is different from Fourier layers in FNO because we only perform one-sided (positive-frequency) operations, whereas FNO performs both positive- and negative-frequency operations. This is because, in AFD, negative frequencies are redundant, as they can be determined by the positive ones via complex conjugation.

We also point out that, since Fourier basis kernel $e^{2\pi j \phi \cdot (\xi - y_i(a))}$ lies in $\mathcal{H}(\mathcal{M})$, which is closed under finite linear combinations, the reproducing kernel $k_a(\xi)$ is guaranteed to lie in $\mathcal{H}(\mathcal{M})$ as well. In addition, although Fourier basis kernels are orthogonal to each other, the reproducing kernels are not. Thus, orthogonalization is still needed.

**Orthogonal reproducing kernels.** Like AFD, in AFDONet, a set of reproducing kernels in Equation 7, each corresponding to one of the $N$ distinct poles $a_1, \ldots, a_N \in \mathcal{M}$, need to be first orthogonalized via Gram-Schmidt orthogonalization:

$$\mathscr{B}_1 = \frac{k_{a_1}(\xi)}{\|k_{a_1}(\xi)\|_{\mathcal{H}(\mathcal{M})}}; \quad \mathscr{B}_i = \frac{k_{a_i}(\xi) - \sum_{j=1}^{i-1} \langle k_{a_i}(\xi), \mathscr{B}_j \rangle \mathscr{B}_j}{\left\| k_{a_i}(\xi) - \sum_{j=1}^{i-1} \langle k_{a_i}(\xi), \mathscr{B}_j \rangle \mathscr{B}_j \right\|_{\mathcal{H}(\mathcal{M})}} \quad \text{for } i = 2, \ldots, N. \tag{8}$$

To adaptively select the poles, we develop a maximum selection principle that is analogous to Equation 3 in AFD theory as:

$$\left| \mathrm{FM}\left(\tilde{u}(x, \cdot)\right) \star \mathscr{B}_i \right| \geq \rho_i \sup\left\{ \left| \mathrm{FM}\left(\tilde{u}(x, \cdot)\right) \star \mathscr{B}_i' \right| : b_i \in \mathcal{M} \backslash \{a_1, \ldots, a_{i-1}\} \right\}, \tag{9}$$

where $\mathscr{B}_1' = \frac{k_{b_1}(\xi)}{\|k_{b_1}(\xi)\|_{\mathcal{H}(\mathcal{M})}}$, $\mathscr{B}_i' = \frac{k_{b_i}(\xi) - \sum_{j=1}^{i-1} \langle k_{b_i}(\xi), \mathscr{B}_j \rangle \mathscr{B}_j}{\left\| k_{b_i}(\xi) - \sum_{j=1}^{i-1} \langle k_{b_i}(\xi), \mathscr{B}_j \rangle \mathscr{B}_j \right\|_{\mathcal{H}(\mathcal{M})}}$ for $i = 2, \ldots, N$, and $k_{b_i}$ is the reproducing kernel at $b_i$.

**The AFD-type decoder network** reconstructs PDE solutions from $\mathrm{FM}\left(\tilde{u}(x, \cdot)\right)$ once the RKHS and its reproducing kernel are established. The decoder adopts a dynamic convolutional kernel network (CKN) (Mairal et al., 2014; Chen et al., 2020b), which (i) performs cross-correlation between $\mathrm{FM}\left(\tilde{u}(x, \cdot)\right)$ and the orthogonal reproducing kernels $\mathscr{B}_i$, (ii) assigns a multiplier $0 < \rho_0 \leq \rho_i < 1$

to the output of each convolutional layer, and (iii) incorporates skip connections for each convolutional layer. With this, the output of the dynamic CKN with $N$ convolutional layers (each pole is associated with a layer) replicates the AFD operation and reconstructs the PDE solution as:

$$\hat{u}_{N,\theta}(x,\cdot) = \sum_{i=1}^{N} \langle \text{FM}\left(\tilde{u}(x,\cdot)\right), \mathscr{B}_{i+\tau_i}\rangle \mathscr{B}_{i+\tau_i} = \sum_{i=1}^{N} \left(\text{FM}\left(\tilde{u}(x,\cdot)\right) \star \mathscr{B}_i\right) \mathscr{B}_{i+\tau_i}, \quad (10)$$

where $\star$ is the cross-correlation defined as $f \star g(\tau_i) = \int_{\mathcal{M}} \bar{f}(z)g(z+\tau_i)\mathrm{d}z$ and $\tau_i$ can choose between $0$ and $N-i$ for convolutional layer $i$.

**Training.** Overall, our AFDONet model is trained end-to-end by minimizing the loss function:

$$\mathcal{L}(\theta) = \underbrace{\left|\left|u(x,\cdot) - \hat{u}_{N,\theta}(x,\cdot)\right|\right|_{\mathcal{H}(\mathcal{M})}^2}_{\text{reconstruction loss in RKHS}} + \underbrace{\left|\left|\tilde{u}(x,\cdot) - \text{FM}(\tilde{u}(x,\cdot))\right|\right|_{H(\mathcal{M})}^2}_{\text{feature map loss}}$$

$$+ \omega \underbrace{\mathcal{D}_{\text{KL}}\Big(\mathcal{CN}(\mu,\sigma^2) \,\Big|\Big|\, \mathcal{CN}(0,I_r)\Big)}_{\text{latent space regularization}} + \underbrace{\sum_{i=0}^{k} w_i \left|\left|\nabla^i \hat{u}_{N,\theta}(x,\cdot) - \nabla^i u(x,\cdot)\right|\right|_{L^2(\mathcal{M})}^2}_{\text{holomorphic training loss}}, \quad (11)$$

where $\nabla^i u$ denotes the $i$-th covariant derivative defined on manifold $\mathcal{M}$. Notice that here, we extend the idea of Sobolev training (Czarnecki et al., 2017) to the complex domain and introduce a holomorphic training loss to enforce consistency with the ground truth solutions both at the function value level and across all orders of derivatives. This enables AFDONet to better capture the inherent smoothness and analytic structure of the target function.

# 6 PROPERTIES OF AFDONET

The design of AFDONet architecture is fully guided by the AFD theory, making it mathematically interpretable in several aspects. Here, we list three important properties of AFDONet:

1. Under the loss function of Equation 11, we can rigorously bound the error of AFDONet in Theorem 1, which is formally stated and proved in Appendix A.

2. By extending the work of Caragea et al. (2022), we can rigorously prove the existence of RKHS $\mathcal{H}(\mathcal{M})$ through the construction of feature map $\text{FM}(\cdot)$ in the latent-to-RKHS network in Theorem 2 (see proof in Appendix B).

3. To ensure convergence of AFDONet, we leverage the convergence mechanism of AFD to design a convergent dynamic CKN decoder by regulating the layer width, depth, and kernel complexity based on the number of samples and the intrinsic smoothness of the target function. This result is formalized in Theorem 3 and is stated and proved in Appendix C.

# 7 EXPERIMENTS

We evaluate the performance of our proposed model across three different PDEs on different manifolds whose solution spaces are not necessarily an RKHS, and compare it with recent neural PDE solvers including FNO (Li et al., 2020; 2023b), WNO (Tripura & Chakraborty, 2023), D-FNO (Li & Ye, 2025), and DeepONet (Lu et al., 2019). Then, we present some key results from selected ablation studies to demonstrate the need for each of the core components of our AFDONet framework. The detailed experimental settings and the complete numerical results can be found in Appendix E. Additional experiments and their results, including one using real-world noisy dataset and another defined on an arbitrary manifold, are discussed in Appendix F.

## 7.1 PDE PROBLEM SETTINGS

**Helmholtz equation on planar manifold with boundary.** Let $(\mathcal{M}, g)$ be a smooth planar Riemannian manifold with boundary $\mathcal{M} \subset \mathbb{R}^2$ equipped with the Euclidean-induced metric $g$. We

consider the 2-D Helmholtz equation on $\mathcal{M}$ with perfectly-matched layer (PML) absorption on $\partial\mathcal{M}$ as follows:

$$\Delta_{\mathcal{M}}u(x,y) \; + \; k^2 n^2(x,y) \, u(x,y) = -\, S(x,y), \quad (x,y) \in \mathcal{M},$$
$$\text{PML absorption on } \partial\mathcal{M}, \tag{12}$$

where wavenumber $k$ is a positive constant, $n : \mathcal{M} \to \mathbb{C}$ is the complex refractive-index field, and $S : \mathcal{M} \to \mathbb{C}$ is the source density. In our experiment, the planar manifold is constructed following Marchand (2023). Furthermore, one can show that the solutions of the Helmholtz equation naturally span an RKHS (see Appendix D).

**Incompressible Navier-Stokes equation on a torus.** Let $(\mathbb{T}^2, g)$ denote a flat two-dimensional torus $\mathbb{T}^2 = \big([0, 2\pi] \times [0, 2\pi]\big)/\sim$ obtained by identifying opposite edges of the square and endowed with the Euclidean metric $g$. It is worth noting that this two-dimensional torus is a compact manifold without boundary, thus it is not diffeomorphic to an open rectangular domain (which is non-compact) or a closed rectangular domain (which has boundary). In other words, even though this flat two-dimensional torus can be projected onto a rectangular domain, it does not necessarily have the same "shape" as a regular domain (e.g., a rectangular domain) from a topological perspective. For viscosity $\nu > 0$, we study the 2-D incompressible Navier-Stokes system:

$$\partial_t \mathbf{u} + (\mathbf{u}\cdot\nabla)\,\mathbf{u} = -\nabla p \; + \; \nu\,\Delta_{\mathbb{T}^2}\mathbf{u}, \qquad (x,y,t) \in \mathbb{T}^2 \times (0,T],$$
$$\nabla_{\mathbb{T}^2}\cdot\mathbf{u} = 0, \qquad\qquad\qquad (x,y,t) \in \mathbb{T}^2 \times [0,T], \tag{13}$$
$$\mathbf{u}(\,\cdot\,,0) = \mathbf{u}_0, \qquad\qquad\qquad x \in \mathbb{T}^2,$$

where $\mathbf{u} = (u,v) : \mathbb{T}^2 \times [0,T] \to \mathbb{R}^2$ is the velocity field and $p : \mathbb{T}^2 \times [0,T] \to \mathbb{R}$ is the pressure.

**Homogeneous Poisson equation on a quarter-cylindrical surface.** Let $(\mathcal{M}, g)$ be a smooth two-dimensional Riemannian manifold $\mathcal{M} = \Big\{(\cos\phi, \sin\phi, z) \in \mathbb{R}^3 \; : \; 0 < \phi < \frac{\pi}{2}, \; 0 < z < L\Big\}$, which restricts the lateral surface of the unit cylinder to a single quadrant. The metric $g$ is the Euclidean metric pulled back by the embedding, so that in local coordinates $(\phi, z)$ one has $\Delta_{\mathcal{M}} = \partial_{\phi\phi} + \partial_{zz}$. We study the 2-D homogeneous Poisson problem with Dirichlet boundary conditions on $\partial\mathcal{M}$:

$$-\Delta_{\mathcal{M}}u(\phi,z) = f(\phi,z), \qquad (\phi,z) \in (0,\tfrac{\pi}{2}) \times (0,L),$$
$$u(\phi,z) = 0, \qquad\qquad (\phi,z) \in \partial\mathcal{M}, \tag{14}$$

where the source term $f(\phi,z) = \beta\Big[\big(\frac{\alpha\pi}{L}\big)^2(1-\cos\phi) - \big(\cos\phi + \sin\phi - 4\sin\phi\cos\phi\big)\Big]\sin\big(\frac{\alpha\pi z}{L}\big)$ (Kamilis, 2013).

Since Helmholtz and Poisson equations are stationary, we focus on the static task for both problems. And for the Navier-Stokes equation, we consider both static and autoregressive tasks.

## 7.2 RESULTS AND DISCUSSIONS

**Comparison with benchmark methods.** In Table 1, we report the performance of AFDONet and benchmark methods in terms of average mean absolute error (MAE) and relative $L^2$ error, as well as their standard deviations ($\pm$) obtained using five random seeds and dataset size of $5000$. Synthetic datasets are generated using finite difference and isogeometric methods, and each model is trained on a 60/20/20 split of training, validation, and testing data. We conclude that, given different dataset sizes, our AFDONet solver consistently outperforms FNO-based solvers and DeepONet across all PDE cases on manifolds. Note that FNO, D-FNO, AFNO, and WNO solvers rely on fast Fourier transform and wavelet transform, both of which are inherently defined on Euclidean domain and thus do not generalize well to curved geometries. Specifically, FNO uses fixed global Fourier bases, which struggle with sharp discontinuities and non-periodic boundaries, and WNO uses fixed wavelets. Meanwhile, DeepONet does not exploit the spectral sparsity of the solution space. In contrast, AFDONet adaptively selects analytic modes and employs pullback operators to ensure accurate, manifold-aware representations. It uses adaptive rational orthogonal bases (i.e., the Takenaka-Malmquist system) parameterized by poles that are learned from input data. This allows the bases to locally adapt to the spatiotemporal dynamics of the solution profile, such as sharp gradients.

Table 1: Average MAE and relative $L^2$ errors and their standard deviations for different PDE benchmark solvers obtained using five random seeds. Dataset size is $5000$. The best results are bolded. All values in the table have been multiplied by $100$.

| Equation | Metric | AFDONet (Ours) | FNO | D-FNO | WNO | DeepONet |
|---|---|---|---|---|---|---|
| Helmholtz 12 | MAE | **0.937 ± 0.063** | 1.855 ± 0.165 | 6.085 ± 0.355 | 11.701 ± 1.429 | 16.224 ± 1.054 |
| | Rel. $L^2$ | **8.141 ± 1.401** | 11.915 ± 0.935 | 39.191 ± 9.361 | 69.735 ± 12.675 | 46.310 ± 10.540 |
| Navier-Stokes (Static) 13 | MAE | **0.332 ± 0.030** | 2.908 ± 0.741 | 0.375 ± 0.103 | 3.974 ± 0.005 | 3.189 ± 0.164 |
| | Rel. $L^2$ | **0.882 ± 0.059** | 7.567 ± 0.173 | 0.996 ± 0.263 | 9.989 ± 0.004 | 7.251 ± 0.422 |
| Navier-Stokes (Autoreg.) 13 | MAE | **0.068 ± 0.037** | 2.386 ± 0.249 | 0.142 ± 0.009 | 3.826 ± 0.191 | 3.168 ± 0.221 |
| | Rel. $L^2$ | **0.170 ± 0.104** | 6.288 ± 0.820 | 0.298 ± 0.060 | 9.541 ± 0.475 | 7.071 ± 0.897 |
| Poisson 14 | MAE | **0.158 ± 0.033** | 0.777 ± 0.093 | 0.343 ± 0.066 | 0.770 ± 0.161 | 0.531 ± 0.030 |
| | Rel. $L^2$ | **0.472 ± 0.109** | 2.567 ± 0.502 | 0.513 ± 0.242 | 1.754 ± 0.943 | 0.483 ± 0.305 |

**Scalability of AFDONet.** In Figure 7.2, we show that AFDONet is scalable subject to increasing dataset size for all benchmark PDE problems considered.

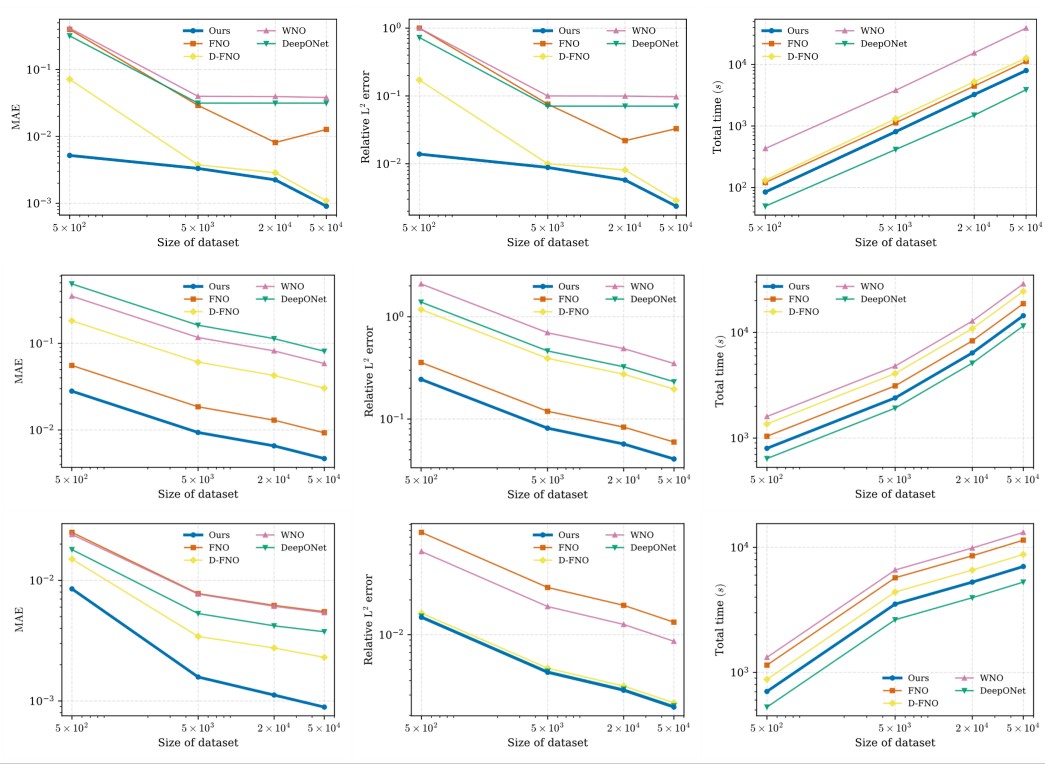

Figure 2: Average MAE, relative $L^2$ error, and total computational time comparisons with respect to dataset size (averaged over five random seeds) for Navier-Stokes equation (static task) (top row), Helmholtz equation (middle row), and Poisson equation (bottom row).

**Computational performance.** From Figure 3, we find empirically that the total time scales almost linearly with respect to the dataset size, as for $Z$ data points, the expensive operations (e.g., the Gram-Schmidt orthogonalization in our decoder) happen $Z$ times per epoch. In addition, we remark that the computational cost for AFDONet per data point is significantly lower than that of FNO. This is because, first, AFDONet operates on a compact latent space (with a dimension of $10$ in our experiments) after the encoder. Thus, the number of basis functions $N$ in the decoder is also small ($N = 3$ in our experiments). Second, the expensive Gram-Schmidt orthogonalization scales with $N^2$, while FNO uses a lifting layer with a width (channel dimension) of $W = 32$ in our setting.

However, in every Fourier layer, FNO performs dense matrix multiplications to mix these channels for every frequency mode. This cost scales with $W^2$, which boils down to $32^2 = 1024$ operations per mode. Last but not least, AFDONet only performs one-sided (positive-frequency) operations due to the nature of AFD, while FNO implements both positive and negative-frequency operations, consuming twice as much memory and computational load.

**Latent-to-RKHS network vs. Latent-to-kernel network.** Our decoder operates within an RKHS $\mathcal{H}(\mathcal{M})$, which is constructed via a latent-to-RKHS network. This network maps latent representations to their nearest RKHS within a Hilbert space. To understand the need for function restrictions within an RKHS, we conduct an ablation study and compare the latent-to-RKHS network with the latent-to-kernel network (Lu et al., 2020), which directly maps latent representations to a kernel function that does not necessarily satisfy the reproducing property. By comparing the results in Tables 1 and 2, we observe that latent-to-RKHS network consistently outperforms the latent-to-kernel network. Both MAE and relative $L^2$ error show at least an order of magnitude reduction for all PDE cases *except* the Helmholtz equation 12, which only yields a slight performance gain. This is due to the fact that the solution space for the Helmholtz equation 12 is already an RKHS (See Appendix D). This illustrates the need and benefit of restricting the latent representations to their RKHS.

Table 2: Ablation studies of our AFDONet architecture show that latent-to-RKHS and AFD-type dynamic CKN decoder work synergistically to improve the solution accuracy. Note that the results for the full architecture are presented in Table 1. The dataset size is 5000.

| Equation | Metric | Latent-to-kernel network + AFD-type decoder | Latent-to-RKHS network + MLP-type decoder | Latent-to-RKHS network + propagation decoder | Latent-to-RKHS + AFD-type decoder (static CNN) | Latent-to-RKHS network + AFD-type decoder (without Equation 9) |
|---|---|---|---|---|---|---|
| Helmholtz 12 | MAE | 1.27E-02 ± 1.91E-03 | 2.11E-01 ± 2.04E-03 | 1.93E-01 ± 5.11E-02 | 2.41E-02 ± 1.16E-02 | 1.81E-01 ± 5.16E-02 |
|  | Rel. $L^2$ | 8.89E-02 ± 6.90E-03 | 1.17 ± 1.22E-02 | 1.07 ± 2.64E-01 | 1.72E-01 ± 9.13E-02 | 1.10 ± 2.62E-01 |
| Navier-Stokes (Static) 13 | MAE | 8.32E-02 ± 1.46E-02 | 4.00E-01 ± 4.46E-03 | 3.98E-01 ± 4.68E-04 | 7.12E-02 ± 1.20E-02 | 1.27E-02 ± 2.03E-03 |
|  | Rel. $L^2$ | 2.19E-01 ± 3.44E-02 | 1.00 ± 9.36E-03 | 1.00 ± 8.30E-06 | 1.85E-01 ± 3.54E-02 | 3.71E-02 ± 6.29E-03 |
| Navier-Stokes (Autoreg.) 13 | MAE | 6.11E-02 ± 2.92E-03 | 1.45E-01 ± 2.59E-02 | 1.48E-01 ± 1.09E-01 | 8.32E-02 ± 9.28E-03 | 2.53E-03 ± 8.26E-04 |
|  | Rel. $L^2$ | 1.58E-01 ± 9.20E-03 | 3.85E-01 ± 6.84E-02 | 3.91E-01 ± 2.30E-01 | 2.16E-01 ± 2.35E-02 | 7.80E-03 ± 1.10E-03 |
| Poisson 14 | MAE | 3.16E-01 ± 8.76E-04 | 1.71E-02 ± 7.73E-03 | 1.81E-02 ± 1.84E-03 | 6.08E-02 ± 6.88E-03 | 3.53E-02 ± 5.51E-03 |
|  | Rel. $L^2$ | 9.77E-01 ± 2.31E-03 | 5.10E-02 ± 2.22E-02 | 5.61E-02 ± 2.17E-02 | 1.77E-01 ± 5.16E-03 | 1.30E-01 ± 1.44E-02 |

**AFD-type decoder vs. other decoder architectures.** We conduct ablation studies by replacing our full AFD-type dynamic CKN decoder with three alternatives, namely an MLP decoder, a propagation decoder (Lu et al., 2020; Buchberger et al., 2020), and an AFD-type decoder with a static CNN. As shown in Table 2, full AFD-type dynamic CKN decoder achieves the best performance for all PDE cases. The improvements are especially significant for the Navier-Stokes equation 13 and Poisson equation 14, where both the MAE and relative $L^2$ error are reduced by one to two orders of magnitude compared to the benchmark decoders. Also, we observe that AFD-type decoder with a static CNN performs slightly worse than our AFD-type dynamic CKN decoder since CNN uses stationary kernels that lack adaptability to the varying spatiotemporal dynamics in PDE solutions. In contrast, dynamic CKN enables data-driven, non-stationary kernel learning, which can better capture these inherent dynamics, especially for heterogeneous equations such as the Poisson equation 14 or time-dependent equations like the Navier-Stokes equation 13.

**Need for VAE backbone.** We design a new ablation study for the Navier-Stokes example with randomized vortex field dataset (see Appendix E.3 for details). The randomized vortex field dataset exhibits sharp gradients and turbulence-like behavior and includes a phase shift for the $v$-component. Therefore, the dynamics of this dataset are challenging to learn. Our goal is to determine whether the $v$-component solution profile would visually match with the ground truth solution when the VAE backbone and its components are removed or replaced. From Table 3, it is clear that the synergistic integration of VAE backbone, latent-to-RKHS network, and AFD-type decoder is essential in accurately capturing $v$-component solution profile in the dataset. Guided by the AFD theory in their design and integration, these components come together to establish the accuracy of our AFDONet solver.

Table 3: Ablation study of replacing VAE with multi-layer fully-connected feedforward (MLP) network as the encoder. Here, ✓: $v$-component solution dynamics visually matches with the ground truth solution; ✗: $v$-component solution dynamics does not visually match with the ground truth.

| Backbone | Full AFDONet (latent-to-RKHS network + AFD-type decoder + Equation 9 | Latent-to-kernel network + AFD-type decoder | Latent-to-RKHS + MLP-type decoder | Latent-to-RKHS + propagation decoder | Latent-to-RKHS + AFD-type decoder (static CNN) | Latent-to-RKHS + AFD-type decoder (without maximal (without Equation 9) |
|---|---|---|---|---|---|---|
| VAE | ✓ | ✗ | ✗ | ✗ | ✓ | ✓ |
| Without VAE (encoder deterministic MLP) | ✗ | ✗ | ✗ | ✗ | ✗ | ✗ |

## 8 CONCLUSION

Existing neural PDE solvers do not perform well to PDEs on manifolds, mainly due to the lack of mathematically grounded methods to design tailored neural network architectures. In this work, we introduce AFDONet, a new neural PDE solver for solving general nonlinear PDEs on smooth manifolds. AFDONet is the first neural PDE solver whose architectural and component design is fully guided by the AFD theory. Thus, it exhibits exceptional mathematical explainability and groundness, and enjoys several desired properties, such as convergence guarantee. AFDONet also achieves outstanding solution accuracy and competitive computational efficiency in benchmark problems studied. In particular, thanks to its deep connections with AFD theory, AFDONet shows superior performance in solving PDEs on i) arbitrary (Riemannian) manifolds, and ii) datasets with sharp gradients. Overall, this work presents a new paradigm for designing explainable neural operator frameworks.

## 9 REPRODUCIBILITY STATEMENT

The source code is uploaded as part of the supplementary material. A complete description of the data processing steps is provided in Appendix E. The assumptions made in proving Theorems 1 through 3 are provided in Appendices A through C, respectively.

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

## A    PROOF OF THEOREM 1

Under the loss function of Equation 11, we can rigorously bound the error of AFDONet in Theorem 1, which states:

**Theorem 1.** *Let $\mathcal{P} \subset \mathbb{R}^d$ be compact and $\{(p_i, u_i)\}_{i=1}^{Z}$ be $Z$ i.i.d. samples with $u_i = F(p_i) + \xi_i$, $\xi_i \sim SubGaussian(\mathcal{H}(\mathcal{M}))$, and $\mathbb{E}[\xi_i] = 0$, where $F : \mathcal{P} \to \mathcal{H}(\mathcal{M})$ is holomorphic, and $\mathcal{H}(\mathcal{M})$ is an RKHS with a kernel $k_m$ whose eigenvalues decay polynomially with rate $k$. Suppose $L_d = \mathcal{O}(\log Z)$ and $W_d = \mathcal{O}(Z^{\frac{1}{2(k+1)}})$ in the decoder network. For the minimizer $\hat{\theta}$ of the loss function $\mathcal{L}(\theta)$ in Equation 11, there exists a constant $C > 0$ such that:*

$$\mathbb{E}\left[\left\|\hat{u}_{N,\hat{\theta}} - F\right\|_{\mathcal{H}(\mathcal{M})}^2\right] \leq CZ^{-\frac{2k+1}{2(k+1)}}(\log Z)^2.$$

We introduce and prove a few lemmas before proving Theorem 1. We assume that the neural network $f_\theta$ is Lipschitz continuous with respect to hyperparameters $\theta$ (i.e., $\|f_\theta - f_{\theta'}\|_{\mathcal{H}} \leq L_f \|\theta - \theta'\|_2$).

**Lemma 1.** *For any $0 < \delta < 1$, for the class of complex-analytic networks with depth $L_d$ and width $W_d$, denoted as $\mathcal{N}_{L_d, W_d, N}$, there exists $\dot{C} > 0$ such that:*

$$\log \mathcal{N}\left(\delta, \mathcal{N}_{L_d, W_d, N}, \|\cdot\|_{\mathcal{H}}\right) \leq \dot{C} W_d L_d \log \left(\frac{W_d L_d}{\delta}\right),$$

*where $\mathcal{N}\left(\delta, \mathcal{N}_{L_d, W_d, N}, \|\cdot\|_{\mathcal{H}}\right)$ means the $\delta$-covering number of $(\mathcal{N}_{L_d, W_d, N}, \|\cdot\|_{\mathcal{H}})$.*

*Proof.* Let us consider the $p$-dimensional $\ell_2$-unit ball $\mathcal{B}^p(1) = \{x \in \mathbb{R}^p : \|x\|_2 \leq 1\}$. Results for covering $\mathcal{B}^p$ (Wainwright, 2019) concludes:

$$\log \mathcal{N}(\delta, \mathcal{B}^p(1), \|\cdot\|_2) \leq p \log \left(1 + \frac{2}{\delta}\right) \leq p \log \left(\frac{3}{\delta}\right). \tag{15}$$

Extending this result to a $\ell_2$-ball of radius $R$, Equation 15 becomes:

$$\log \mathcal{N}(\delta, \mathcal{B}^p(R), \|\cdot\|_2) \leq p \log \left(1 + \frac{2R}{\delta}\right) \leq p \log \left(\frac{3R}{\delta}\right) \tag{16}$$

by rescaling $\delta$ in the RHS of Equation 15 with $\delta/R$. Furthermore, by letting $p = 2W_d L_d$, Equation 16 becomes:

$$\log \mathcal{N}(\delta, \mathcal{B}^{2W_d L_d}(R), \|\cdot\|_2) \leq 2W_d L_d \log \left(\frac{3R}{\delta}\right). \tag{17}$$

From the Lipschitz property and the fact that the parameter space of $\mathcal{N}_{L_d, W_d, N}$ can be controlled by $\mathcal{B}^{2W_d L_d}(R)$, we have:

$$\log \mathcal{N}\left(\delta, \mathcal{N}_{L_d, W_d, N}, \|\cdot\|_{\mathcal{H}}\right) \leq \log \mathcal{N}\left(\frac{\delta}{L_f}, \mathcal{B}^{2W_d L_d}(R), \|\cdot\|_2\right) \leq 2W_d L_d \log \left(\frac{3L_f R}{\delta}\right), \tag{18}$$

where $L_f$ is the Lipschitz constant. With $R = \mathcal{O}(W_d L_d)$, Equation 18 leads to:

$$\log \mathcal{N}\left(\delta, \mathcal{N}_{L_d, W_d, N}, \|\cdot\|_{\mathcal{H}}\right) \leq \dot{C} W_d L_d \log \left(\frac{W_d L_d}{\delta}\right), \tag{19}$$

which completes the proof. $\qquad\square$

**Lemma 2.** *For $a > 1$ and $0 < r \leq \min(a, e)$ where $e$ is the base of the natural logarithm, there exists $b > 0$ that satisfies the following inequality:*

$$r\sqrt{\log\left(\frac{a}{r}\right)} \leq \sqrt{b}\sqrt{r \log a}.$$

*Proof.* For the case $1 < r \leq \min(a, e)$, we may choose $b = e$. Squaring both sides of the inequality and rearranging lead to $(r - e) \log a \leq r \log r$. Suppose $r = e$, the inequality is automatically satisfied for any $a > 1$. Suppose $r < e$, since $a \geq r$, we have: $(r - e) \log a \leq (r - e) \log r$. Thus, it suffices to show $(r - e) \log r \leq r \log r$, which is equivalent to showing $e \log r \geq 0$. This is automatically satisfied because $0 < \log r \leq 1$.

For the case $0 < r \leq 1$, we rearrange the inequality and obtain $b \geq \frac{r(\log a - \log r)}{\log a} > 0$. Furthermore, $\frac{r(\log a - \log r)}{\log a}$ reaches its maximum, $\frac{a}{e \log a}$, at $r = \frac{a}{e}$. Thus, suppose $a \leq e$, then we may choose $b \geq \frac{a}{e \log a}$ and the inequality is satisfied. Suppose $a \geq e$, then $\max \frac{r(\log a - \log r)}{\log a} = 1$ within $0 < r \leq 1$. Thus, we may choose $b \geq 1$ and the inequality is satisfied. $\qquad\square$

**Lemma 3.** *There exists $\widetilde{C} > 0$ such that:*

$$\mathbb{E}_{\epsilon}\left[\sup_{f \in \mathscr{F}} \left|\frac{1}{Z}\sum_{i=1}^{Z} \epsilon_i f_\theta(p_i)\right|\right] \leq \widetilde{C}\sqrt{\frac{r W_d L_d \log(W_d L_d)}{Z}},$$

*where $\epsilon_i$ are i.i.d. Rademacher variables and $\mathscr{F}$ is a function class for a radius $0 < r \leq e$ defined as $\{f \in \mathcal{N}_{L_d, W_d, N} : \|f - F\|_{\mathcal{H}} \leq r\}$.*

*Proof.* From Dudley's entropy integral bound (Wainwright, 2019), we have:

$$\mathbb{E}_\epsilon \left[ \sup_{f \in \mathscr{F}} \left| \frac{1}{Z} \sum_{i=1}^{Z} \epsilon_i f(p_i) \right| \right] \leq \frac{24}{\sqrt{Z}} \int_\varepsilon^{2r} \sqrt{\log \mathcal{N}(t, \mathscr{F}, \| \cdot \|_\mathcal{H})} dt. \tag{20}$$

Since $\mathcal{N}(\delta, \mathscr{F}, \| \cdot \|_\mathcal{H}) \leq \mathcal{N}(\delta, \mathcal{N}_{L_d, W_d, N}, \| \cdot \|_\mathcal{H})$ and according to Lemma 1, Equation 20 becomes:

$$\begin{aligned} \mathbb{E}_\epsilon \left[ \sup_{f \in \mathscr{F}} \left| \frac{1}{Z} \sum_{i=1}^{Z} \epsilon_i f(p_i) \right| \right] &\leq \frac{24}{\sqrt{Z}} \int_\varepsilon^{2r} \sqrt{\log \mathcal{N}(t, \mathcal{N}_{L_d, W_d, N}, \| \cdot \|_\mathcal{H})} \, dt \\ &\leq \frac{24}{\sqrt{Z}} \int_\varepsilon^{2r} \sqrt{\dot{C} W_d L_d \log \left( \frac{W_d L_d}{t} \right)} \, dt. \end{aligned} \tag{21}$$

To evaluate the integral on the RHS of Equation 21, we apply the change of variables technique by defining $u = \log \left( \frac{W_d L_d}{t} \right)$ (and thus $dt = -W_d L_d e^{-u} du$):

$$\begin{aligned} \int_\varepsilon^{2r} \sqrt{\log \left( \frac{W_d L_d}{t} \right)} dt &= \int_{\log(\frac{W_d L_d}{2r})}^{\log(\frac{W_d L_d}{\varepsilon})} \sqrt{u} \cdot W_d L_d e^{-u} \, du \\ &= W_d L_d \left[ \Gamma \left( \frac{3}{2}, \log \left( \frac{W_d L_d}{2r} \right) \right) - \Gamma \left( \frac{3}{2}, \log \left( \frac{W_d L_d}{\varepsilon} \right) \right) \right] \\ &= 2r \sqrt{\log \left( \frac{W_d L_d}{2r} \right)} + \mathcal{O} \left( \frac{r}{\log(\frac{W_d L_d}{2r})} \right), \end{aligned} \tag{22}$$

where $\Gamma(s, x) = \int_x^\infty t^{s-1} e^{-t} dt$ is the upper incomplete gamma function.

Substituting Equation 22 into Equation 21 and applying Lemma 2 lead to:

$$\begin{aligned} \mathbb{E}_\epsilon \left[ \sup_{f \in \mathscr{F}} \left| \frac{1}{Z} \sum_{i=1}^{Z} \epsilon_i f(p_i) \right| \right] &\leq 24 \cdot 2r \sqrt{\frac{\dot{C} W_d L_d \log \left( \frac{W_d L_d}{2r} \right)}{Z}} \\ &\leq 24 \sqrt{b} \sqrt{\frac{2r \dot{C} W_d L_d \log (W_d L_d)}{Z}} \\ &\leq \widetilde{C} \sqrt{\frac{r W_d L_d \log (W_d L_d)}{Z}}, \end{aligned} \tag{23}$$

where $\widetilde{C} \geq 24 \sqrt{2b\dot{C}}$. $\qquad\square$

**Lemma 4.** *Let $\hat{\theta}$ minimize the loss function $\mathcal{L}$ in Equation 11. With probability at least $1 - e^{-t}$ for all $t \geq 0$,*

$$\mathcal{L}(\hat{\theta}) \leq \inf_\theta \mathcal{L}(\theta) + \hat{C} \frac{W_d L_d \log(W_d L_d) + t}{Z}$$

*holds for some $\hat{C}$.*

*Proof.* From the symmetrization inequality (Boucheron et al., 2012), we have:

$$\mathbb{E} \left[ \mathcal{L}(\hat{\theta}) - \mathcal{L}(\theta) \right] \leq 2 \mathbb{E} \left[ \sup_{f \in \mathscr{F}} \frac{1}{Z} \sum_{i=1}^{Z} \epsilon_i f(p_i) \right], \tag{24}$$

where $\epsilon_i$ are i.i.d. Rademacher variables.

Let us define the centered process:

$$\mathscr{Z} = \sup_{f \in \mathscr{F}} \sum_{i=1}^{Z} (f(p_i) - \mathbb{E}[f(p_i)]) \tag{25}$$

under the assumptions that there exists $\mathscr{Z}'_k$ such that: (i) $\mathscr{Z}'_k \leq \mathscr{Z} - \mathscr{Z}_k \leq 1$ almost surely; (ii) $\mathbb{E}^k[\mathscr{Z}'_k] \geq 0$, where $\mathbb{E}^k$ is the expectation taken conditionally to the sigma field generated by

$(p_1, \ldots, p_{k-1}, p_{k+1}, \ldots p_Z)$; and (iii) there exists $q > 0$ such that $\mathscr{Z}'_k \leq q$ almost surely. Here, $\mathscr{Z}_k = \sup_{f \in \mathscr{F}} \sum_{i \neq k} (f(p_i) - \mathbb{E}[f(p_i)])$.

Applying Bennett concentration inequality (Bousquet, 2002) to the process $\mathscr{Z}$ leads to:

$$\mathbb{P}\left( \mathscr{Z} \geq \mathbb{E}[\mathscr{Z}] + \sqrt{2vt} + \frac{t}{3} \right) \leq e^{-t}, \tag{26}$$

where $v = (1 + q)\mathbb{E}[\mathscr{Z}] + Z\sigma^2$ and $\sigma^2 \geq \frac{1}{\mathscr{Z}} \sum_{k=1}^{Z} \mathbb{E}^k \left[ (\mathscr{Z}'_k)^2 \right]$.

Combining Equations 24, 26 and 26 with probability at least $1 - e^{-t}$, we have:

$$\mathcal{L}(\hat{\theta}) - \mathcal{L}(\theta) \leq 2\mathbb{E}\left[ \sup_{f \in \mathscr{F}} \frac{1}{Z} \sum_{i=1}^{Z} \epsilon_i f(p_i) \right] + \frac{1}{Z}\left( \sqrt{2vt} + \frac{t}{3} \right). \tag{27}$$

Moreover, by putting $\mathbb{E}_\epsilon \left[ \sup_{f \in \mathscr{F}} \left| \frac{1}{Z} \sum_{i=1}^{Z} \epsilon_i f(p_i) \right| \right] \asymp r$ for Lemma 3 ($\asymp$ stands for asymptotic equivalence), we obtain:

$$r \asymp \frac{W_d L_d \log(W_d L_d)}{Z}. \tag{28}$$

Extending the result of Equation 24 to $\mathscr{Z}$ defined in Equation 25 leads to:

$$\begin{aligned} \mathbb{E}[\mathscr{Z}] &\leq 2Z\mathbb{E}\left[ \sup_{f \in \mathscr{F}} \frac{1}{Z} \sum_{i=1}^{Z} \epsilon_i f(p_i) \right] \\ &\leq 2Z\widetilde{C}\sqrt{\frac{rW_d L_d \log(W_d L_d)}{Z}} \asymp 2\widetilde{C}W_d L_d \log(W_d L_d), \end{aligned} \tag{29}$$

where the second inequality and last asymptotic equivalence come from Lemma 3 and Equation 28, respectively.

According to Efron-Stein inequality (Boucheron et al., 2012), there exists $\mathscr{Z}'_k = \mathscr{Z} - \mathscr{Z}_k$, such that:

$$\sigma^2 \leq \sum_{k=1}^{Z} \mathbb{E}\left[ (\mathscr{Z} - \mathbb{E}[\mathscr{Z} \mid p_k])^2 \right] \leq \mathbb{E}^k[(\mathscr{Z}'_k)^2], \tag{30}$$

where $\mathscr{Z} \mid p_k$ excludes $p_k$ from $\mathscr{Z}$. Thus, to derive an upper bound on $\mathbb{E}^k[(\mathscr{Z}'_k)^2]$, we write:

$$(\mathscr{Z}'_k)^2 \leq \left( \sup_{f \in \mathscr{F}} |f(p_k) - \mathbb{E}[f(p_k)]| \right)^2 \leq 2\left( \sup_{f \in \mathscr{F}} f(p_k)^2 + \mathbb{E}[f(p_k)]^2 \right) \leq 4\sup_{f \in \mathscr{F}} f(p_k)^2, \tag{31}$$

where the second inequality comes from $(a - b)^2 \leq 2(a^2 + b^2)$ and the last inequality holds by Jensen's inequality ($\mathbb{E}[f(p_k)]^2 \leq \mathbb{E}[f(p_k)^2]$). Then, for $f \in \mathscr{F}$ and a bounded function $F$, it follows:

$$\mathbb{E}[f(p_k)^2] \leq 2(\|f - F\|_{\mathcal{H}}^2 + \|F\|_{\mathcal{H}}^2) \leq 2(r^2 + \|F\|_{\mathcal{H}}^2). \tag{32}$$

Substituting the result of Equation 32 into Equation 31 and combining it with Equation 30 give:

$$\sigma^2 \leq Dr^2 \asymp \left( \frac{W_d L_d \log(W_d L_d)}{Z} \right)^2, \tag{33}$$

for some $D > 0$.

Substituting Equations 33 and 29 into 26 gives:

$$\begin{aligned} v = (1 + q)\mathbb{E}[\mathscr{Z}] + Z\sigma^2 &\leq C'(1 + q)W_d L_d \log(W_d L_d) + \frac{(W_d L_d \log(W_d L_d))^2}{Z} \\ &\leq \left( C'(1 + q) + \frac{1}{Z} \right)(W_d L_d \log(W_d L_d))^2. \end{aligned} \tag{34}$$

Substituting Equations 34 and 28 into 27 gives:

$$\mathcal{L}(\hat{\theta}) \leq \mathcal{L}(\theta) + 2\widetilde{C}\frac{W_d L_d \log(W_d L_d)}{Z} + \sqrt{2\left[C'(1+q) + \frac{1}{Z}\right]t}\frac{W_d L_d \log(W_d L_d)}{Z} + \frac{t}{3Z} \quad (35)$$

$$\leq \mathcal{L}(\theta) + \hat{C}\frac{W_d L_d \log(W_d L_d) + t}{Z}$$

holds for any $\theta$, where $\hat{C} = \max\left\{2\widetilde{C}, \sqrt{2\left[C'(1+q) + \frac{1}{Z}\right]t}, \frac{1}{3}\right\}$. Thus, we conclude that $\mathcal{L}(\hat{\theta}) \leq \inf_\theta \mathcal{L}(\theta) + \hat{C}\frac{W_d L_d \log(W_d L_d) + t}{Z}$. $\qquad\square$

PROOF OF THEOREM 1

*Proof.* From Lemma 4, we know that with probability at least $1 - e^{-t}$ for all $t \geq 0$ and some $\hat{C}$,

$$\mathcal{L}(\hat{\theta}) \leq \inf_\theta \mathcal{L}(\theta) + \hat{C}\frac{W_d L_d \log(W_d L_d) + t}{Z}. \quad (36)$$

Realizing $\mathcal{L}(\theta) \asymp \|\hat{u}_{N,\theta} - F\|_{\mathcal{H}}^2$, then for $s_0 = \inf_\theta \mathcal{L}(\theta) + \hat{C}\frac{W_d L_d \log(W_d L_D) + t_0}{Z}$, it holds that:

$$\begin{aligned}
\mathbb{E}[\mathcal{L}(\hat{\theta})] &\leq \int_0^\infty \mathbb{P}(\mathcal{L}(\hat{\theta}) \geq s)ds \\
&= \int_0^{s_0} \mathbb{P}(\mathcal{L}(\hat{\theta}) \geq s)ds + \int_{s_0}^\infty \mathbb{P}(\mathcal{L}(\hat{\theta}) \geq s)ds \\
&\leq s_0 + M \cdot e^{-t_0} \\
&= s_0 + \frac{M}{Z},
\end{aligned} \quad (37)$$

where $t_0 = \log Z$ and we assume that $\mathcal{L} \leq M$ for $t > t_0$.

Since $L_d = \mathcal{O}(\log Z)$ and $W_d = \mathcal{O}(Z^{\frac{1}{2(k+1)}})$, we have:

$$\begin{aligned}
\frac{W_d L_d \log(W_d L_d)}{Z} &\asymp \frac{Z^{\frac{1}{2(k+1)}} \cdot \log Z \cdot \log(Z^{\frac{1}{2(k+1)}} \log Z)}{Z} \\
&= \frac{Z^{\frac{1}{2(k+1)}} \cdot \log Z \cdot \left(\frac{1}{2(k+1)} \log Z + \log\log Z\right)}{Z} \\
&\asymp Z^{\frac{1}{2(k+1)} - 1} \cdot \log Z \cdot \log Z \\
&= Z^{-\frac{2k+1}{2(k+1)}}(\log Z)^2.
\end{aligned} \quad (38)$$

Combining Equations 36, 37 and 38 leads to the final result:

$$\mathbb{E}\left[\left\|\hat{u}_{N,\hat{\theta}} - F\right\|_{\mathcal{H}}^2\right] \leq C Z^{-\frac{2k+1}{2(k+1)}}(\log Z)^2 + \mathcal{O}(Z^{-1}), \quad (39)$$

where $C > 0$ is a constant and the term $\mathcal{O}(Z^{-1})$ vanishes for a large $Z$. $\qquad\square$

## B  PROOF OF THEOREM 2

**Theorem 2.** *Let $H$ be a Hilbert space on a manifold $\mathcal{M}$. Fix $d, n \in \mathbb{N}$, then for any $\widetilde{x} \in H(\mathcal{M})$ and any $\varepsilon > 0$, there exist a convolutional kernel $K$ defining an RKHS $\mathcal{H}(\mathcal{M})$ and a complex-valued modReLU neural network $\mathrm{FM}_{\theta'}$ with at most $C\ln(2/\varepsilon)$ layers, $C\eta^{-2d/n}\ln^2(2/\varepsilon)$ weights, and weights bounded by $C\varepsilon^{-44d}$ such that*

$$\mathrm{FM}_{\theta'}(\widetilde{x}) \in \mathcal{H}(\mathcal{M}) \quad \text{and} \quad \|\widetilde{x} - \mathrm{FM}_{\theta'}(\widetilde{x})\|_{H(\mathcal{M})} \leq \inf_\theta \|\widetilde{x} - \mathrm{FM}_\theta(\widetilde{x})\|_{H(\mathcal{M})} + \varepsilon,$$

*where $C = C(d, n) > 0$ depends only on the dimension $d$ and the smoothness parameter $n$.*

*Proof.* First, we show that $\mathcal{H}(\mathcal{M})$ exists by introducing a map $\Phi : H(\mathcal{M}) \rightarrow \mathcal{H}(\mathcal{M})$ and the reproducing kernel is defined as $K(x, x') = \langle \Phi(x), \Phi(x') \rangle_{\mathcal{H}(\mathcal{M})}$. Specifically, the map $\Phi(x)$ corresponding to a convolutional kernel $K$ can be represented as $\mathcal{A}_L \circ \mathcal{M}_L \circ \mathcal{P}_L \cdots \mathcal{A}_1 \circ \mathcal{M}_1 \circ \mathcal{P}_1 x$ where $L$ is the depth of the kernel and $\mathcal{A}_l, \mathcal{M}_l$ and $\mathcal{P}_l$ are the linear operators related to pooling, kernel mapping and patch extraction, respectively (Bietti, 2022). Without loss of generality, we assume that $\mathcal{H}(\mathcal{M}) \subset H(\mathcal{M})$. Next, we point out that $\mathcal{H}(\mathcal{M})$ is convex by showing that, for any two functions $f, g \in \mathcal{H}(\mathcal{M})$:

$$
\begin{aligned}
\alpha f + (1 - \alpha)g = \alpha \langle f, \mathcal{A}_L \circ \mathcal{M}_L \circ \mathcal{P}_L \cdots \mathcal{A}_1 \circ \mathcal{M}_1 \circ \mathcal{P}_1 x \rangle_{\mathcal{H}(\mathcal{M})} + (1 - \alpha) \\
\langle g, \mathcal{A}_L \circ \mathcal{M}_L \circ \mathcal{P}_L \cdots \mathcal{A}_1 \circ \mathcal{M}_1 \circ \mathcal{P}_1 x \rangle_{\mathcal{H}(\mathcal{M})} \\
= \langle \alpha f + (1 - \alpha)g, \mathcal{A}_L \circ \mathcal{M}_L \circ \mathcal{P}_L \cdots \mathcal{A}_1 \circ \mathcal{M}_1 \circ \mathcal{P}_1 x \rangle_{\mathcal{H}(\mathcal{M})}
\end{aligned}
\tag{40}
$$

for $\alpha \in [0, 1]$. Thus, $\mathcal{H}(\mathcal{M})$ is closed due to the closedness of manifold $\mathcal{M}$ and the completeness of Hilbert space $\mathcal{H}$.

Next, from the Hilbert projection theorem, for $\widetilde{x} \in H(\mathcal{M})$, there exists a unique $y \in \mathcal{H}(\mathcal{M})$ such that, for any $\widetilde{y} \in \mathcal{H}(\mathcal{M})$, $||\widetilde{x} - y||_{H(\mathcal{M})} \leq ||\widetilde{x} - \widetilde{y}||_{H(\mathcal{M})}$. Let us denote $y$ as $\Psi(\widetilde{x})$, where $\Psi$ is a map from $H(\mathcal{M})$ to $\mathcal{H}(\mathcal{M})$. Following the main result of Caragea et al. (2022), for any $\widetilde{y} \in \mathcal{H}(\mathcal{M})$ and any $\varepsilon > 0$, there exists a complex-valued modReLU neural network with hyperparameters $\theta$, $\text{FM}_\theta$, containing no more than $C \ln(2/\varepsilon)$ layers, $C\eta^{-2d/n} \ln^2(2/\varepsilon)$ weights (all weights bounded by $C\varepsilon^{-44d}$), such that $||\widetilde{y} - \text{FM}_\theta(\widetilde{x})||_{H(\mathcal{M})} < \frac{\varepsilon}{2}$. In addition, there also exists another complex-valued modReLU neural network with hyperparameters $\theta'$, $\text{FM}_{\theta'}$, such that $||\Psi(\widetilde{x}) - \text{FM}_{\theta'}(\widetilde{x})||_{H(\mathcal{M})} < \frac{\varepsilon}{2}$. Thus, we have:

$$
\begin{aligned}
||\widetilde{x} - \text{FM}_{\theta'}(\widetilde{x})||_{H(\mathcal{M})} &= ||\widetilde{x} - \Psi(\widetilde{x}) + \Psi(\widetilde{x}) - \text{FM}_{\theta'}(\widetilde{x})||_{H(\mathcal{M})} \\
&\leq ||\widetilde{x} - \Psi(\widetilde{x})||_{H(\mathcal{M})} + ||\Psi(\widetilde{x}) - \text{FM}_{\theta'}(\widetilde{x})||_{H(\mathcal{M})} \\
&\leq ||\widetilde{x} - \widetilde{y}||_{H(\mathcal{M})} + \frac{\varepsilon}{2} \\
&= ||\widetilde{x} - \widetilde{y} + \text{FM}_\theta(\widetilde{x}) - \text{FM}_\theta(\widetilde{x})||_{H(\mathcal{M})} + \frac{\varepsilon}{2} \\
&\leq ||\widetilde{x} - \text{FM}_\theta(\widetilde{x})||_{H(\mathcal{M})} + ||\text{FM}_\theta(\widetilde{x}) - \widetilde{y}||_{H(\mathcal{M})} + \frac{\varepsilon}{2} \\
&\leq ||\widetilde{x} - \text{FM}_\theta(\widetilde{x})||_{H(\mathcal{M})} + \varepsilon.
\end{aligned}
\tag{41}
$$

This completes the proof. $\qquad\square$

## C  PROOF OF THEOREM 3

**Theorem 3.** *Let $L_d$, $W_d$, and $N$ denote the depth, width, and number of layers of dynamic CKN decoder network satisfying Equation 9. For any $\varepsilon > 0$, there exist $L_d = \mathcal{O}\left(\log \frac{1}{\varepsilon}\right), W_d = \mathcal{O}\left(\varepsilon^{-\frac{1}{k+1}}\right), N = \mathcal{O}\left(\log \frac{1}{\varepsilon}\right)$ and $\theta \in \mathcal{N}_{L_d, W_d, N}$ such that*

$$
\sup_{p \in \mathcal{P}} ||\hat{u}_{N,\theta} - F(p)||_{\mathcal{H}(\mathcal{M})} \leq \varepsilon,
$$

*where $\mathcal{N}_{L_d, W_d, N}$ is the class of complex-analytic networks with depth $L_d$ and width $W_d$.*

To prove Theorem 3, we first introduce and/or prove a few lemmas.

**Lemma 5** ((Yarotsky, 2017)). *For any dimension $n$, smoothness parameter $k + 1$, and error tolerance $\varepsilon \in (0, 1)$, there exists a ReLU neural network architecture such that it can approximate any function $f$ with accuracy $\varepsilon$, i.e., with approximation error at most $\varepsilon$. The network has depth at most $c(\ln(1/\varepsilon) + 1)$, and uses at most $c\varepsilon^{-\frac{d}{n}}(\ln(1/\varepsilon) + 1)$ weights and computation units, where $c = c(d, n)$ is a constant depending only on $d$ and $n$.*

**Lemma 6.** *Let $f \in C^k([0, 1]^d)$ or $W^{k+1, \infty}([0, 1]^d)$, for $\varepsilon > 0$, there exists a ReLU network $f_\theta$ with width $W_d = \mathcal{O}\left(\varepsilon^{-\frac{d}{k+1}}\right)$ such that $||f - f_\theta||_{L^\infty} \leq \varepsilon$.*

*Proof.* The result follows from Lemma 5, which states that for any $d \in \mathbb{N}$, $n \in \mathbb{N}$, and $\varepsilon \in (0, 1)$, there exists a ReLU neural network of depth $\mathcal{O}(\log(1/\varepsilon))$ and size $\mathcal{O}(\varepsilon^{-\frac{d}{n}} \log(1/\varepsilon))$ that can uniformly approximate any function in the class $F_{d,n}$, which includes functions in $W^{n, \infty}([0, 1]^d)$ with

bounded norm. By setting $n = k + 1$, it holds that $f \in W^{k+1,\infty}([0, 1]^d)$, with the network width scaling as $\mathcal{O}(\varepsilon^{-\frac{d}{k+1}})$, up to a logarithmic factor. Note that any $f \in C^k([0, 1]^d)$ with bounded derivatives up to order $k$ also belongs to $W^{k,\infty}([0, 1]^d)$ and can be embedded into $W^{k+1,\infty}$. Thus, Lemma 6 holds for any $f \in C^k([0, 1]^d)$. $\square$

**Remark.** *The result of Lemma 6 is nearly optimal. Yarotsky (2017, Theorem 5) shows that there exist functions $f \in W^{n,\infty}([0, 1]^d)$ for which the complexity $N(f, \varepsilon)$ is not $o(\varepsilon^{-\frac{d}{9n}})$ as $\varepsilon \to 0$. This implies that no network architecture can uniformly approximate all such functions with significantly better scaling in $\varepsilon$.*

**Lemma 7.** *Let $\mathcal{H}$ be a separable Hilbert space and $f \in \mathcal{H}$ belong to a class of functions with $k$-th order smoothness. For $\varepsilon > 0$, there exists a ReLU network $f_\theta$ with width $W_d = \mathcal{O}\left(\varepsilon^{-\frac{d}{k+1}}\right)$ such that $\|f - f_\theta\|_{\mathcal{H}} \leq \varepsilon$.*

*Proof.* Assume $f \in \text{dom}(A^{-k})$ with respect to its operator $A$ with input dimension $d$. Let $\{e_j\}_{j=1}^{\infty}$ be an orthonormal basis of $\mathcal{H}$ with associated eigenvalues $\lambda_j \asymp j^{2\alpha}$ (assuming that $\alpha \geq \frac{k+1}{2dk}$) of $A$. Then, we have $\|A^k f\|_{\mathcal{H}}^2 = \sum_{j=1}^{\infty} \lambda_j^{2k}|\langle f, e_j\rangle|^2 < \infty$. We can define the eigenexpansion of $f$ as $P_N f = \sum_{j=1}^{N}\langle f, e_j\rangle e_j$ and $\|f - P_N f\|_{\mathcal{H}} \leq CN^{-(k+\frac{1}{2})\alpha} \leq \varepsilon/2$ holds for $N = \lceil \varepsilon^{-\frac{1}{2\alpha k + \alpha}} \rceil \asymp \varepsilon^{-\frac{1}{2k\alpha}}$. In the finite-dimensional subspace $\text{span}\{e_1, ..., e_N\} \cong \mathbb{R}^N$, each coordinate function $f_j = \langle f, e_j\rangle$ inherits $C^k$ regularity and can be approximated by a ReLU network $\tilde{f}_j$ with $|\tilde{f}_j(x) - f_j(x)| \leq \frac{\varepsilon}{2\sqrt{N}}$ using width $\mathcal{O}(\varepsilon^{-\frac{d}{k+1}})$ per coordinate from Lemma 6. The RELU network $f_\theta = \sum_{j=1}^{N} \tilde{f}_j e_j$ then satisfies $\|f - f_\theta\|_{\mathcal{H}} \leq \|f - P_N f\|_{\mathcal{H}} + \sqrt{\sum_j \|\tilde{f}_j - f_j\|_{L^\infty}^2} \leq \varepsilon$. The total width $W_d = \mathcal{O}(N \cdot \varepsilon^{-\frac{d}{k+1}}) = \mathcal{O}(\varepsilon^{-\frac{d}{k+1}})$ $\square$

PROOF OF THEOREM 3

*Proof.* First, we show that, for a sufficiently large $N$ and any $\varepsilon > 0$,

$$\|\hat{u}_{N,\theta} - \text{FM}(\tilde{u})\|_{\mathcal{H}(\mathcal{M})} \leq \frac{\varepsilon}{4} \tag{42}$$

holds. From Equation 10, we have $\hat{u}_{N,\theta} = \sum_{i=1}^{N}\langle \text{FM}(\tilde{u}), \mathcal{B}_{i+\tau_i}\rangle \mathcal{B}_{i+\tau_i}$. Here, we prove by contradiction. Suppose $\|\hat{u}_{N,\theta} - \text{FM}(\tilde{u})\|_{\mathcal{H}(\mathcal{M})} > \frac{\varepsilon}{4}$, then there exists an open ball $\mathcal{B}$ and $C > 0$ such that:

$$\left\|\text{FM}(\tilde{u}(x, \cdot)) - \sum_{i=1}^{N}\langle \text{FM}(\tilde{u}(x, \cdot)), \mathcal{B}_{i+\tau_i}\rangle \mathcal{B}_{i+\tau_i}\right\|_{\mathcal{H}(\mathcal{M})} = C\max_{m,\xi}(\|k_m(\xi)\|) > \frac{\varepsilon}{4}, \tag{43}$$

for $(x, \cdot) \in \mathcal{B} \subset \mathcal{M}$. Furthermore, since the term $\sum_{i=1}^{N}\|\langle \text{FM}(\tilde{u}(x, \cdot)), \mathcal{B}_{i+\tau_i}\rangle\|_{\mathcal{H}(\mathcal{M})}^2 < \infty$ is finite, there exists $N_0$ such that for any $n \geq N_0$, we have:

$$\sum_{i=n}^{N}\|\langle \text{FM}(\tilde{u}(x, \cdot)), \mathcal{B}_{i+\tau_i}\rangle\|_{\mathcal{H}(\mathcal{M})}^2 < \left(\frac{\rho_0 C}{2}\right)^2. \tag{44}$$

Next, we examine the term $\|\langle u_n, \frac{k_b}{\|k_b\|}\rangle\|_{\mathcal{H}(\mathcal{M})}$, where $(x, b) \in \mathcal{B}$ and

$$
\begin{aligned}
u_n &= \text{FM}(\tilde{u}(x, \cdot)) - \sum_{i=1}^{n-1}\langle \text{FM}(\tilde{u}(x, \cdot)), \mathcal{B}_{i+\tau_i}\rangle \mathcal{B}_{i+\tau_i} \\
&= \text{FM}(\tilde{u}(x, \cdot)) - \sum_{i=1}^{N}\langle \text{FM}(\tilde{u}(x, \cdot)), \mathcal{B}_{i+\tau_i}\rangle \mathcal{B}_{i+\tau_i} + \sum_{i=n}^{N}\langle \text{FM}(\tilde{u}(x, \cdot)), \mathcal{B}_{i+\tau_i}\rangle \mathcal{B}_{i+\tau_i}.
\end{aligned} \tag{45}
$$

Therefore, we have:

$$
\begin{aligned}
\left\| \langle u_n, \frac{k_b}{\|k_b\|} \rangle \right\|_{\mathcal{H}(\mathcal{M})} &= \left\| \left\langle \mathrm{FM}\,(\tilde{u}) - \sum_{i=1}^{N} \langle \mathrm{FM}\,(\tilde{u})\,, \mathscr{B}_{i+\tau_i} \rangle \mathscr{B}_{i+\tau_i} + \sum_{i=n}^{N} \langle \mathrm{FM}\,(\tilde{u})\,, \mathscr{B}_{i+\tau_i} \rangle \mathscr{B}_{i+\tau_i}, \frac{k_b}{\|k_b\|} \right\rangle \right\|_{\mathcal{H}(\mathcal{M})} \\
&\geq \left\| \left\langle \mathrm{FM}\,(\tilde{u}) - \sum_{i=1}^{N} \langle \mathrm{FM}\,(\tilde{u})\,, \mathscr{B}_{i+\tau_i} \rangle \mathscr{B}_{i+\tau_i}, \frac{k_b}{\|k_b\|} \right\rangle \right\|_{\mathcal{H}(\mathcal{M})} \\
&\quad - \left\| \left\langle \sum_{i=n}^{N} \langle \mathrm{FM}\,(\tilde{u})\,, \mathscr{B}_{i+\tau_i} \rangle \mathscr{B}_{i+\tau_i}, \frac{k_b}{\|k_b\|} \right\rangle \right\|_{\mathcal{H}(\mathcal{M})} \\
&\geq \left\| \frac{\left( \mathrm{FM}\,(\tilde{u}) - \sum_{i=1}^{N} \langle \mathrm{FM}\,(\tilde{u})\,, \mathscr{B}_{i+\tau_i} \rangle \mathscr{B}_{i+\tau_i} \right)\big|_b}{\|k_b\|} \right\|_{\mathcal{H}(\mathcal{M})} \\
&\quad - \sqrt{\sum_{i=n}^{N} \| \langle \mathrm{FM}\,(\tilde{u}(x,\cdot))\,, \mathscr{B}_{i+\tau_i} \rangle \|_{\mathcal{H}(\mathcal{M})}^2} \\
&\geq C - \frac{C}{2} = \frac{C}{2},
\end{aligned}
\tag{46}
$$

where the third inequality holds due to the reproducing property of RKHS: $\langle f, k_m \rangle = f(m)$.

Meanwhile, there exists $\gamma > 0$ satisfying Equation 9 such that:

$$
\begin{aligned}
\left\| \langle u_n, \frac{k_b}{\|k_b\|} \rangle \right\|_{\mathcal{H}(\mathcal{M})} &= \frac{\left\| \langle u_n, k_b - \sum_{i=1}^{n-1} \langle k_b, \mathscr{B}_{i+\tau_i} \rangle \mathscr{B}_{i+\tau_i} \rangle \right\|_{\mathcal{H}(\mathcal{M})}}{\|k_b\|} \\
&\leq \frac{\left\| \langle u_n, k_b - \sum_{i=1}^{n-1} \langle k_b, \mathscr{B}_{i+\tau_i} \rangle \mathscr{B}_{i+\tau_i} \rangle \right\|_{\mathcal{H}(\mathcal{M})}}{\left\| k_b - \sum_{i=1}^{n-1} \langle k_b, \mathscr{B}_{i+\tau_i} \rangle \mathscr{B}_{i+\tau_i} \right\|_{\mathcal{H}(\mathcal{M})}} \\
&= \left\| \langle u_n, \mathscr{B}_{n+\tau_n}^b \rangle \right\|_{\mathcal{H}(\mathcal{M})} \\
&\leq \left\| \frac{1}{\rho_0} \langle u_n, \mathscr{B}_{n+\tau_n} \rangle - \frac{\gamma}{\rho_0} \right\|_{\mathcal{H}(\mathcal{M})} \\
&\leq \frac{1}{\rho_0} \cdot \frac{\rho_0 C}{2} - \frac{\gamma}{\rho_0} \\
&< \frac{C}{2}.
\end{aligned}
\tag{47}
$$

Hence, Equations 46 and 47 lead to a contradiction. Therefore, Equation 42 must hold.

Next, from Theorem 2, there exists a network FM with appropriate hyperparameters $\theta'$ such that:

$$
\|\tilde{u} - \mathrm{FM}_{\theta'}(\tilde{u})\|_{H(\mathcal{M})} \leq \inf_{\theta} \|\tilde{u} - \mathrm{FM}_{\theta}(\tilde{u})\|_{H(\mathcal{M})} + \frac{\varepsilon}{4}.
\tag{48}
$$

Let us denote $\mathrm{FM}_{\theta'}$ as FM. Note that $\tilde{u}$ in Equation 48 lies in the Hilbert space $H(\mathcal{M})$, not the RKHS $\mathcal{H}(\mathcal{M})$. Furthermore, from Lemma 5, there exists a set of hyperparameters $\widetilde{\theta}$ such that $\|\tilde{u} - \mathrm{FM}_{\widetilde{\theta}}(\tilde{u})\|_{H(\mathcal{M})} \leq \frac{\varepsilon}{4}$. Therefore, Equation 48 reduces to:

$$
\begin{aligned}
\|\tilde{u} - \mathrm{FM}(\tilde{u})\|_{H(\mathcal{M})} &\leq \inf_{\theta} \|\tilde{u} - \mathrm{FM}_{\theta}(\tilde{u})\|_{H(\mathcal{M})} + \frac{\varepsilon}{4} \\
&\leq \|\tilde{u} - \mathrm{FM}_{\widetilde{\theta}}(\tilde{u})\|_{H(\mathcal{M})} + \frac{\varepsilon}{4} \\
&\leq \frac{\varepsilon}{4} + \frac{\varepsilon}{4} = \frac{\varepsilon}{2}.
\end{aligned}
\tag{49}
$$

From Lemma 7, for $\widetilde{u}$ which is the output of a neural network with width $W_d = \mathcal{O}\left(\varepsilon^{-\frac{d}{k+1}}\right)$, we have:

$$\|\widetilde{u} - F\|_{H(\mathcal{M})} \leq \frac{\varepsilon}{4}. \tag{50}$$

Putting Equations 42, 49 and 50 together leads to:

$$\begin{aligned}
\|\hat{u}_{N,\theta} - F(p)\|_{\mathcal{H}(\mathcal{M})} &\leq \|\hat{u}_{N,\theta} - \mathrm{FM}\left(\tilde{u}\right)\|_{\mathcal{H}(\mathcal{M})} + \|\widetilde{u} - \mathrm{FM}(\widetilde{u})\|_{H(\mathcal{M})} + \|\widetilde{u} - F\|_{H(\mathcal{M})} \\
&\leq \varepsilon
\end{aligned} \tag{51}$$

for any $p \in \mathcal{P}$. Therefore, taking supremum on LHS and RHS of Equation 51, we have proven Theorem 3. $\qquad\square$

## D    PROOF THAT THE HELMHOLTZ EQUATION SPANS AN RKHS

Let us consider the Helmholtz equation $\Delta_{\mathcal{M}} u + k^2 u = 0$ without loss of generality. We first introduce some background and preliminaries before proceeding with the proof.

Let $\Delta = \sum_{i=1}^{n} \frac{\partial^2}{\partial x_i^2}$ be the Euclidean Laplace operator acting on the Sobolev space of weakly twice differentiable functions defined on $\mathbb{R}^n$. Let $k > 0$ be a fixed constant. A function $u$ defined on $\mathbb{R}^n$ is called a solution of the Helmholtz equation, if $\Delta u + k^2 u = 0$ on $\mathbb{R}^n$. In other words, $u$ satisfies one of the following:

- $u \in C^2(\mathbb{R}^n)$ is a classical solution of the above equation on $\mathbb{R}^n$; or
- $u \in W^2(\mathbb{R}^n)$ is a solution in the weak $L^2$-sense, i.e., $u$ is locally square integrable, and satisfies $\int_{\mathbb{R}^n} u(x) \left[ \Delta\varphi(x) + k^2\varphi(x) \right] dx = 0$ for any (test) function $\varphi \in C^\infty(\mathbb{R}^n)$ with compact support.

It follows from Axler et al. (2001) that any solution of homogeneous Helmholtz equation is real analytic on $\mathbb{R}^n$. We define the following space:

$$W_{\mathrm{Helm},k}(\mathbb{R}^n) = \{u \in C^\infty(\mathbb{R}^n) \mid \Delta u + k^2 u = 0 \text{ on } \mathbb{R}^n\}. \tag{52}$$

Hartman & Wilcox (1961) introduced the concept of Herglotz wave function. The Herglotz wave functions consists of all the entire solutions $u$ of the homogeneous Helmholtz equation $\Delta u + k^2 u = 0$ on $\mathbb{R}^n$ with $k > 0$ such that Herglotz boundedness condition:

$$\lim_{R \to +\infty} \frac{1}{R} \int_{\|x\| < R} |u(x)|^2 \, dx < +\infty \tag{53}$$

holds. Hartman & Wilcox (1961) characterized the Herglotz wave functions as the entire solutions $u$ of the homogeneous Helmholtz equation with far-field pattern in $L^2(\mathbb{S}^{n-1})$. That is, functions $u$ defined on $\mathbb{R}^n$ can be written as:

$$u(x) = \int_{\mathbb{S}^{n-1}} e^{ik\langle x, \xi \rangle} g(\xi) \, d\sigma(\xi), \tag{54}$$

for some $g \in L^2(\mathbb{S}^{n-1})$.

With this, let us consider the Helmholtz equation on the standard $n$-dimensional unit sphere $\mathbb{S}^n = \{x \in \mathbb{R}^{n+1} : \|x\| = 1\}$ in $\mathbb{R}^{n+1}$ with canonical spherical Riemannian metric $g$. Let $\Delta_{\mathbb{S}^n}$ be the spherical Laplacian acting on the Sobolev space $W^2(\mathbb{S}^n)$ of real-valued, square-integrable, and twice weakly differentiable functions on $\mathbb{S}^n$. Consider the Helmholtz equation on the Riemannian manifold $(\mathbb{S}^{n-1}, g)$ with canonical spherical metric $g$. Its entire solution can be expressed as:

$$u = W\phi(x) = (2\pi)^{\frac{1-n}{2}} \int_{\mathbb{S}^{n-1}} e^{ikx \cdot \xi} \phi(\xi) d\sigma(\xi), \tag{55}$$

where $W$ is the Fourier extension operator and $\phi \in L^2(\mathbb{S}^{n-1})$ is Herglotz wave function. It has been shown that $W$ defined in Equation 55 is an isomorphism of $L^2(\mathbb{S}^{n-1})$ onto the space $W^2$ consisting of all solutions of Helmholtz equation with radial and angular derivatives satisfying:

$$||u||^2 = \int_{|x|>1} (|u(x)|^2 + |\frac{\partial u}{\partial r}(x)|^2 + |\frac{\partial u}{\partial \theta}(x)|^2) \frac{dx}{|x|^3} < \infty, \tag{56}$$

(see (Pérez-Esteva & Valenzuela-Díaz, 2017)). In this sense, the space $W^2$ in $\mathbb{R}^2$ is a Hilbert space with reproducing kernel (i.e., RKHS).

Meanwhile, to the best of our knowledge, there exists no such formal analysis on Helmholtz equation on any smooth (Riemannian) manifold $(\mathcal{M}, g)$. For any smooth manifold $(\mathcal{M}, g)$, the Laplace-Beltrami operator $\Delta_{\mathcal{M}}$, defined in Equation **??**, has orthonormal eigenbases on $L^2(\partial\mathcal{M})$ as $\{\psi_\lambda\}_\lambda$ with corresponding eigenvalues $\lambda \geq 0$. For each $\psi_\lambda$, let us consider:

$$(\Delta_{\mathcal{M}} + k^2)\phi_\lambda = 0 \text{ in } \mathcal{M}, \quad \phi_\lambda|_{\partial\mathcal{M}} = \psi_\lambda. \tag{57}$$

By elliptic regularity, $\phi_\lambda \in H^2(\mathcal{M})$. Furthermore, we extend the Fourier extension operator in Equation 55 to $W_{\mathcal{M}}$ on any smooth manifold $\mathcal{M}$:

$$W_{\mathcal{M}}f(x) = \int_{\partial\mathcal{M}} \Psi(x, \xi)f(\xi)d\sigma(\xi), \quad \text{where } \Psi(x, \xi) = \sum_\lambda \phi_\lambda(x)\overline{\psi_\lambda(\xi)}. \tag{58}$$

Now, we present the main result in Theorem 4 that $W^2(\mathcal{M})$ is the space of all Herlotz wave functions.

**Theorem 4.** *The operator $W_{\mathcal{M}} : L^2(\partial\mathcal{M}) \to W^2(\mathcal{M})$ defined in Equation 58 is a topological isomorphism, where $W^2(\mathcal{M}) = \{u \in H^2(\mathcal{M}) : (\Delta_{\mathcal{M}} + k^2)u = 0\}$.*

**Remark.** *Theorem 4 implies that $W_{\mathcal{M}}$ is an isomorphism between $L^2(\partial\mathcal{M})$ and $W^2(\mathcal{M})$, the space of $H^2$-solutions to the Helmholtz equation $(\Delta_{\mathcal{M}} + k^2)u = 0$. Such an isomorphism $W_{\mathcal{M}}$ implies that $\mathcal{H}(\mathcal{M})$ inherits a Hilbert space or RKHS structure from $L^2(\partial\mathcal{M})$. In other words, $W^2(\mathcal{M})$ is an RKHS.*

To prove Theorem 4, we first introduce and prove a lemma.

**Lemma 8.** *Let $J_\nu(z)$ be the Bessel function of order $\nu \in \mathbb{R}$. For each eigenfunction $\psi_j$ of $\Delta_{\partial\mathcal{M}}$, define $F_j = W_{\mathcal{M}}\psi_j$. Then:*

1. *$F_j(x) = (2\pi)^{1/2}i^{\nu(j)}r^{-\frac{n-2}{2}}J_{\nu(j)}(kr)\psi_j(\xi)$, where $x = r\xi$ in normal coordinates near $\partial\mathcal{M}$.*

2. *The family $\{F_j\}$ is orthogonal in $W^2(\mathcal{M})$, and*

$$\|F_j\|_{H^2(\mathcal{M})} = \sqrt{2} + \mathcal{O}\left(\frac{1}{\lambda_j}\right).$$

3. *For $f = \sum_j a_j\psi_j \in L^2(\partial\mathcal{M})$ and $u = \sum_j a_j F_j \in W^2(\mathcal{M})$,*

$$\|u\|_{H^2(\mathcal{M})} \sim \|f\|_{L^2(\partial\mathcal{M})},$$

*with absolute and uniform convergence on compact subsets of $\mathcal{M}$.*

*Proof.* We prove the three components of Lemma 8 as follows:

1. Helmholtz equation $(\Delta_{\mathcal{M}} + k^2)\phi_j = 0$ can be written as:

$$\left(\partial_r^2 + \frac{n-1}{r}\partial_r + \frac{1}{r^2}\Delta_{\partial\mathcal{M}} + k^2\right)(r^{-\frac{n-2}{2}}R_j(r)\psi_j(\xi)) = 0. \tag{59}$$

Substituting $\phi_j = r^{-\frac{n-2}{2}}R_j(r)\psi_j(\xi)$ into Equation 59 yields:

$$R_j'' + \frac{1}{r}R_j' + \left(k^2 - \frac{\nu(j)^2}{r^2}\right)R_j = 0, \tag{60}$$

whose solution is $R_j(r) = J_{\nu(j)}(kr)$. By the Funk-Hecke formula (Xu, 2000), we have:

$$F_j(x) = \int_{\partial\mathcal{M}} \Psi(x, \xi)\psi_j(\xi)d\sigma(\xi) = (2\pi)^{1/2}i^{\nu(j)}r^{-\frac{n-2}{2}}J_{\nu(j)}(kr)\psi_j(\xi). \tag{61}$$

2. Since $\psi_j$ and $\psi_k$ are orthonormal eigenbases, $\psi_j$ and $\psi_k$ are orthogonal on $\partial\mathcal{M}$. Therefore,

$$\langle F_j, F_k \rangle_{H^2(\mathcal{M})} = \int_M \left( \phi_j \overline{\phi_k} + \nabla\phi_j \cdot \overline{\nabla\phi_k} \right) dV_g = 0 \tag{62}$$

for any $j \neq k$. Using the asymptotic $J_{\nu(j)}(kr) \sim \frac{(kr/2)^{\nu(j)}}{\Gamma(\nu(j)+1)}$ for $r \to 0^+$ and oscillatory decay for $r \to \infty$, we have:

$$\|F_j\|_{H^2(\mathcal{M})}^2 = 2 + \mathcal{O}\left(\frac{1}{\lambda_j}\right),$$

where the error term comes from the next-order Bessel asymptotics.

3. From Part 2, the map $f \mapsto u$ is bounded:

$$\|u\|_{H^2(\mathcal{M})}^2 = \sum_j |a_j|^2 \|F_j\|_{H^2(\mathcal{M})}^2 \sim \sum_j |a_j|^2 = \|f\|_{L^2(\partial\mathcal{M})}^2. \tag{63}$$

Next, we prove $|J_\nu(kr)| \sim \mathcal{O}(\nu^{-1/2})$ uniformly holds on compact subsets $K \subset \mathcal{M}$. According to Watson (1922, §8.4), we have:

$$J_\nu(\nu \sec\beta) \sim \left(\frac{2}{\pi\nu\tan\beta}\right)^{1/2} \left[ \cos\left(\nu\tan\beta - \nu\beta - \frac{\pi}{4}\right) \sum_{m=0}^{\infty} \frac{(-1)^m \Gamma(2m+\frac{1}{2})}{\Gamma(\frac{1}{2})} \cdot \right.$$

$$\frac{A_{2m}}{(\frac{1}{2}\nu\tan\beta)^{2m}} + \sin\left(\nu\tan\beta - \nu\beta - \frac{\pi}{4}\right) \sum_{m=0}^{\infty} \frac{(-1)^m \Gamma(2m+\frac{3}{2})}{\Gamma(\frac{1}{2})} \cdot$$

$$\left. \frac{A_{2m+1}}{(\frac{1}{2}\nu\tan\beta)^{2m+1}} \right], \tag{64}$$

where $A_k$ is defined following $A_0 = 1$, $A_1 = \frac{1}{3} + \frac{5}{24}\cot^2\beta$, $A_2 = \frac{3}{128} + \frac{77}{576}\cot^2\beta + \frac{385}{3456}\cot^4\beta$, and so on.

Let $z = \sec\beta$, which implies $\tan\beta = \sqrt{z^2-1}$ and $\cot\beta = \frac{1}{\sqrt{z^2-1}}$. Moreover, $\eta$ is defined as $\eta(z) = \tan\beta - \beta = \sqrt{z^2-1} - \sec^{-1}z$. Then, by $\cos\theta = \Re(e^{i\theta})$, $\sin\theta = \Im(e^{i\theta})$, we have:

$$\cos(\nu\eta - \pi/4) \cdot S_0 + \sin(\nu\eta - \pi/4) \cdot S_1 = \Re\left[e^{i(\nu\eta - \pi/4)}(S_0 - iS_1)\right], \tag{65}$$

where $S_0 = \sum_{m=0}^{\infty} \frac{(-1)^m \Gamma(2m+\frac{1}{2})}{\Gamma(\frac{1}{2})} \cdot \frac{A_{2m}}{(\frac{1}{2}\nu\tan\beta)^{2m}}$ and $S_1 = \sum_{m=0}^{\infty} \frac{(-1)^m \Gamma(2m+\frac{3}{2})}{\Gamma(\frac{1}{2})} \cdot \frac{A_{2m+1}}{(\frac{1}{2}\nu\tan\beta)^{2m+1}}$.

We say that there exists $U_k(p)$ which is a polynomial combination of $A_k$ by comparing $\frac{A_{2m}}{(\nu\tan\beta)^{2m}}$ and $\frac{U_k(p)}{\nu^k}$. By $\tan\beta = \sqrt{z^2-1}$ and $p = \frac{1}{\sqrt{1+z^2}}$, we have:

$$\left(\frac{2}{\pi\nu\tan\beta}\right)^{1/2} = \frac{1}{(1+z^2)^{1/4}} \cdot \frac{1}{\sqrt{2\pi\nu}} \cdot \left(\frac{2z^2}{z^2-1}\right)^{1/4}. \tag{66}$$

Combining Equation 64, Equation 65, and Equation 66 leads to:

$$J_\nu(\nu z) \sim \frac{\exp\left(\nu\eta - \frac{\pi}{4}\right)}{(1+z^2)^{1/4}\sqrt{2\pi\nu}} \left[\sum_{k=0}^{\infty} \frac{U_k(p)}{\nu^k}\right]. \tag{67}$$

Next, for $\nu \gg 1$ and $r \in K$ (i.e., $z = \frac{kr}{\nu}$ is bounded), we have:

$$J_\nu(kr) \approx \left(\frac{2}{\pi\nu}\right)^{1/2} \frac{\cos\left(\nu\eta(z) - \frac{\pi}{4}\right)}{(1+z^2)^{1/4}}. \tag{68}$$

Since $|\cos(\cdot)| \leq 1$ and $(1+z^2)^{1/4}$ has positive lower bound $G$ on $K$, we have:

$$|J_\nu(kr)| \leq G\left(\frac{2}{\pi\nu}\right)^{1/2} = \mathcal{O}(\nu^{-1/2}). \tag{69}$$

Finally, substituting Equation 69 into 63, we have, for compact subsets $K \subset \mathcal{M}$:

$$\sum_j |a_j||F_j(x)| \leq \left(\sum_j |a_j|^2\right)^{1/2} \left(\sum_j |J_{\nu(j)}(kr)|^2\right)^{1/2} < \infty. \tag{70}$$

This completes the proof. □

PROOF OF THEOREM 4

*Proof.* For $f = \sum_j a_j \psi_j \in L^2(\partial\mathcal{M})$, let us define:

$$W_{\mathcal{M}} f = \sum_j a_j F_j, \quad \text{where } F_j = W_{\mathcal{M}} \psi_j. \tag{71}$$

From Part 3 of Lemma 8, the series converges absolutely and uniformly on compact subsets $K$ as:

$$\sum_j |a_j| \|F_j\|_{L^\infty(K)} \leq C \left(\sum_j |a_j|^2\right)^{1/2} \left(\sum_j \lambda_j^{-1/2}\right)^{1/2} < \infty, \tag{72}$$

where $\|F_j\|_{L^\infty(K)} \leq C\lambda_j^{-\frac{1}{4}}$ comes from Bessel decay (Matviyenko, 1993) and $\lambda_j \sim j^{\frac{2}{n-1}}$ comes from Weyl's law (Liokumovich et al., 2018).

Then, from Part 2 of Lemma 8:

$$\|W_{\mathcal{M}} f\|_{H^2(\mathcal{M})}^2 = \sum_j |a_j|^2 \|F_j\|_{H^2(\mathcal{M})}^2 \sim \sum_j |a_j|^2 = \|f\|_{L^2(\partial\mathcal{M})}^2. \tag{73}$$

Next, we prove the surjectivity of $W_{\mathcal{M}}$. Let $u \in W^2(\mathcal{M})$. On $\partial\mathcal{M}$, we expand $u$ in eigenfunctions using:

$$u(r, \xi) = \sum_j A_j(r)\psi_j(\xi), \quad A_j(r) = \langle u(r, \cdot), \psi_j \rangle_{L^2(\partial\mathcal{M})}. \tag{74}$$

This way, the Helmholtz equation $(\Delta_{\mathcal{M}} + k^2)u = 0$ reduces to an ordinary differential equation:

$$A_j'' + \frac{n-1}{r}A_j' + \left(k^2 - \frac{\lambda_j + (\frac{n-2}{2})^2}{r^2}\right)A_j = 0, \tag{75}$$

whose solution is $A_j(r) = a_j r^{-\frac{n-2}{2}} J_{\nu(j)}(kr)$, where $\nu(j) = \sqrt{\lambda_j + (\frac{n-2}{2})^2}$. Therefore, $u = \sum_j a_j F_j = W_{\mathcal{M}} f$ for $f = \sum_j a_j \psi_j \in L^2(\partial\mathcal{M})$. Finally, the inverse $W_{\mathcal{M}}^{-1} : u \mapsto u|_{\partial\mathcal{M}}$ is bounded by the trace theorem (Adams & Fournier, 2003):

$$\|W_{\mathcal{M}}^{-1} u\|_{L^2(\partial\mathcal{M})} = \|u|_{\partial\mathcal{M}}\|_{L^2(\partial\mathcal{M})} \leq C\|u\|_{H^2(\mathcal{M})}. \tag{76}$$

This completes the proof. □

# E  EXPERIMENT DETAILS

In this section, we provide a detailed description of datasets, implementation details, and additional experimental results.

## E.1  DATASETS

**Helmholtz equation.** We generate the dataset using the Helmholtz equation solver `helmhurts-python`, which is available in Marchand (2023). This solver computes the electric field distribution $u(x, y)$ for given $n(x, y)$ and source terms $S(x, y)$, discretized on a uniform grid with resolution $\Delta x = \Delta y = 1\,\text{cm}$. $S(x, y)$ is constructed by assigning a complex-valued excitation $P \cdot e^{i\phi}$ to all pixels marked as sources (RGB (255,0,0)) in the input image, where $P$ is the transmitter power and $\phi = 0$ denotes a uniform phase alignment. Perfectly matched layers (PMLs) of thickness 12 cells absorb outgoing waves to approximate open boundary conditions. We select randomized physical parameters to generate the full dataset, including transmitter power $P \sim \mathcal{U}(0.5, 2.0)$, frequency $f \sim \mathcal{U}(1.5, 3.0)\,\text{GHz}$, and wall properties $\eta \sim \mathcal{U}(1.5, 3.0)$, $\kappa \sim \mathcal{U}(0.05, 0.2)$. The resulting field intensities $|u|$ are log-scaled and normalized to $[0, 1]$.

**Navier-Stokes equation.** The dataset is generated by numerically solving the 2D incompressible Navier-Stokes equations using a spectral method solver adapted from the `NSsimulation` repository (lavenderses, 2021) on a torus. The viscosity $\nu$ are sampled following $\nu \sim \mathcal{U}(0.001, 0.1)$. For the static task, the dataset contains the value of parameters $\alpha$ and the numerical solutions **u**. For the autoregressive task, the dataset contains the numerical solutions $\mathbf{u}(x, t)$ and $\mathbf{u}(x, t+1)$.

**Poisson equation.** Using isogeometric analysis with NURBS basis functions of order $p = 2$ proposed in (Kamilis, 2013), we generate the dataset for this problem by specifying $\alpha \sim (2, 6)$.

E.2 IMPLEMENTATION DETAILS

We run all experiments in a Dell Precision 7920 Tower equipped with Intel Xeon Gold 6246R CPU and NVIDIA Quadro RTX 6000 GPU (with 24GB GGDR6 memory).

In our implementation, the ground-truth PDE solutions $u(x, \cdot)$ are generated as discrete numerical solutions on a grid via finite difference or isogeometric analysis (IGA) depending on the PDE. For the Helmholtz and Navier-Stokes equations, the derivatives $\nabla^i u$ are computed by applying finite-difference schemes (e.g., central difference) to the discrete ground-truth solutions. For the Poisson equation, since the solutions are generated using IGA with NURBS basis functions, we exploit the fundamental property of IGA (Hughes et al., 2005; Piegl & Tiller, 1997): the basis functions possess high-order smoothness, allowing the derivatives of the ground-truth solution to be computed analytically from the NURBS control points and weights, bypassing numerical stability issues. To prevent the instability of training stage, we introduce the weights $\omega_i$ for higher-order derivatives, which are set to $10^{-8}$ in our experiments. Such a small weight can prevent the higher-order derivatives from dominating the loss function in the earlier training stage, thus ensuring stability and guiding AFDONet to capture smoothness and analytic information during training.

For FNO-based solvers (Li et al., 2020; 2023b; Li & Ye, 2025), the number of Fourier modes considered in the spectral convolutions is an important hyperparameter. We find that no more than 16 Fourier modes are enough to solve the three benchmark PDE problems. In fact, increasing the number of Fourier modes beyond 16 could lead to worse performance. From Figure 3, we plot the average MAE and total computational time of FNO with $8, 12, 16, 32, 64, 128$ Fourier modes. As a result, in our experiments, we set the number of Fourier modes to be 12 for all FNO and D-FNO models. Similar trends happen to other benchmark PDE problems, so we use 12 Fourier modes in all benchmark PDE problems.

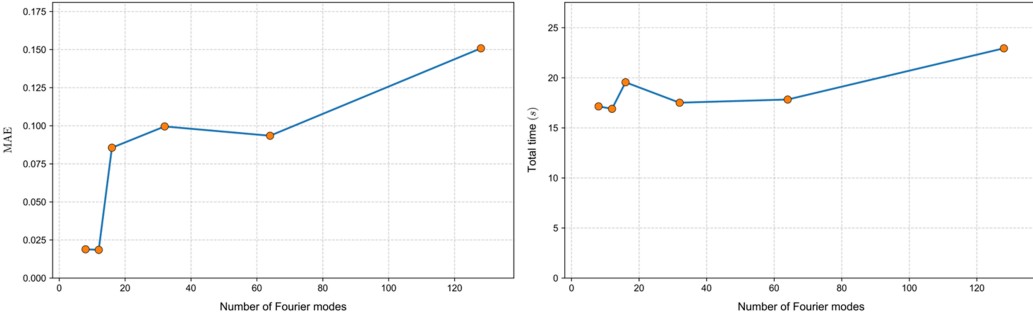

Figure 3: Average MAE and total computational time (in seconds) of FNO solver with respect to number of Fourier modes (averaged over five random seeds) for solving the Helmholtz equation 12.

In addition, for AFDONet, increasing the dimension of the latent space helps achieve higher accuracy. However, this also comes with an increase in computational costs. This is illustrated in Table 4 below taking Navier-Stokes equation. Therefore, to demonstrate the effectiveness of our AFDONet solver even in the worst-case scenario, we set the latent space dimension to 10 for all benchmark PDE problems.

The AFDONet loss function and training specifications are listed in Table 5 below.

For the benchmark solvers, their detailed architectures are as follows:

Table 4: Average MAE, relative $L^2$ error, and computational time (in seconds) of AFDONet (averaged over five random seeds) for solving Navier-Stokes equation 13 (autoregressive task) under different latent space dimensions.

| Latent dimension | MAE | Relative $L^2$ error | Time (sec) |
|---|---|---|---|
| 16 | 6.40E-04 $\pm$ 9.90E-05 | 1.11E-03 $\pm$ 1.91E-04 | 1058.39 $\pm$ 19.30 |
| 20 | 5.35E-04 $\pm$ 1.36E-04 | 1.40E-03 $\pm$ 1.03E-03 | 1190.61 $\pm$ 15.67 |
| 32 | 3.77E-04 $\pm$ 1.28E-04 | 9.60E-04 $\pm$ 8.03E-04 | 1110.57 $\pm$ 18.38 |
| 64 | 4.62E-04 $\pm$ 1.35E-04 | 1.22E-03 $\pm$ 8.92E-04 | 1173.40 $\pm$ 17.22 |
| 100 | 4.05E-04 $\pm$ 1.09E-04 | 1.06E-03 $\pm$ 9.94E-04 | 1365.03 $\pm$ 21.89 |
| 128 | 3.89E-04 $\pm$ 1.26E-04 | 9.99E-04 $\pm$ 8.48E-04 | 1406.05 $\pm$ 23.98 |
| 256 | 5.03E-04 $\pm$ 1.98E-04 | 1.27E-03 $\pm$ 1.14E-03 | 1743.28 $\pm$ 27.64 |

Table 5: Specifications of loss function and training for AFDONet solver.

| Parameter | Value |
|---|---|
| Training epochs | 100 |
| Loss weights ($\omega$) | $10^{-5}$ |
| Loss weights ($w_i$) | $10^{-8}$ |
| Optimizer | Adam |
| Learning rate | $10^{-3}$ |
| Batch size | 16 |
| Encoder hidden layers dimension | 256 |
| Latent space dimension | 10 |

- The FNO solver (Li et al., 2020; 2023b) consists of an initial linear projection layer $P$ (width is 32) followed by 5 Fourier layers with 12 Fourier modes and GeLU activation function. A neural network with two fully connected layers $Q$ (the first layer has 128 neurons and the second layer has 2 neurons) is used to project back to the target dimension. The Adam optimizer (learning rate: $10^{-3}$) is used to train the FNO solver based on minimizing the MSE loss.

- The D-FNO solver (Li & Ye, 2025) has a similar architecture as the FNO solver, except that a reduction layer is introduced between the initial linear projection layer $P$ and the 5 Fourier layers to decompose the output of $P$ into a series of two one-dimensional vectors. The reduction layer does not use traditional neurons. Instead, it projects inputs into a rank-16 subspace via factor matrices (see Equation 6 of Li & Ye (2025)). The Fourier layers have 12 Fourier modes (also suggested by Li & Ye (2025)) and use GeLU activation function. After that, an operation called product is used to put the two vectors together. In D-FNO, $Q$ has two layers (the first layer has 128 neurons and the second layer has one neuron). The Adam optimizer (learning rate: $10^{-3}$) is used to train the D-FNO solver based on minimizing the MSE loss.

- The WNO solver (Tripura & Chakraborty, 2023) adopts the FNO architecture by replacing Fourier layers with wavelet integral layers that decompose the inputs using Daubechies wavelets and apply learnable linear transformations to the wavelet coefficients before reconstruction. The structure of $Q$ is the same as that of FNO. GeLU activation function and the Adam optimizer (learning rate: $10^{-3}$) are used.

- The DeepONet solver (Lu et al., 2019) consists of two subnetworks: a branch network and a trunk network. The branch network which handles the high-dimensional input functions has three fully-connected layers with 64 neurons per layer. The truck network which handles spatial coordinates also has three fully-connected layers with 64 neurons per layer. Their outputs are combined via a dot product. ReLU activation function is employed in both branch and truck networks. We use the Adam optimizer (learning rate: $10^{-3}$) to minimize the MSE loss.

### E.3 ADDITIONAL EXPERIMENTAL RESULTS

VISUALIZATION OF SOLVER PERFORMANCE IN BENCHMARK PDE PROBLEMS

In Figures 4 through 6, we plot the ground truth and predicted solutions of AFDONet and baseline methods for the three case studies. The corresponding MAE and relative $L^2$ error results are listed in Table 1.

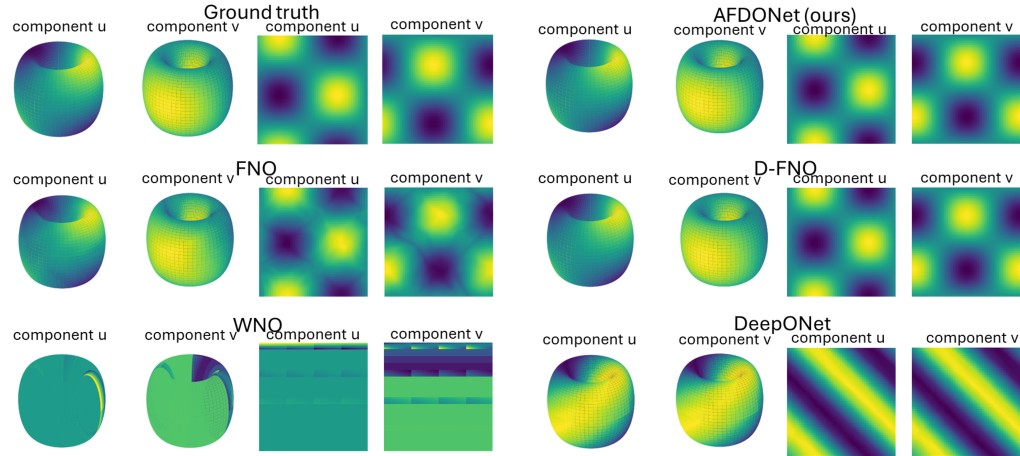

Figure 4: Ground truth and predicted solutions $(u, v)$ of the Navier-Stokes equation (static task) on the torus and heat map.

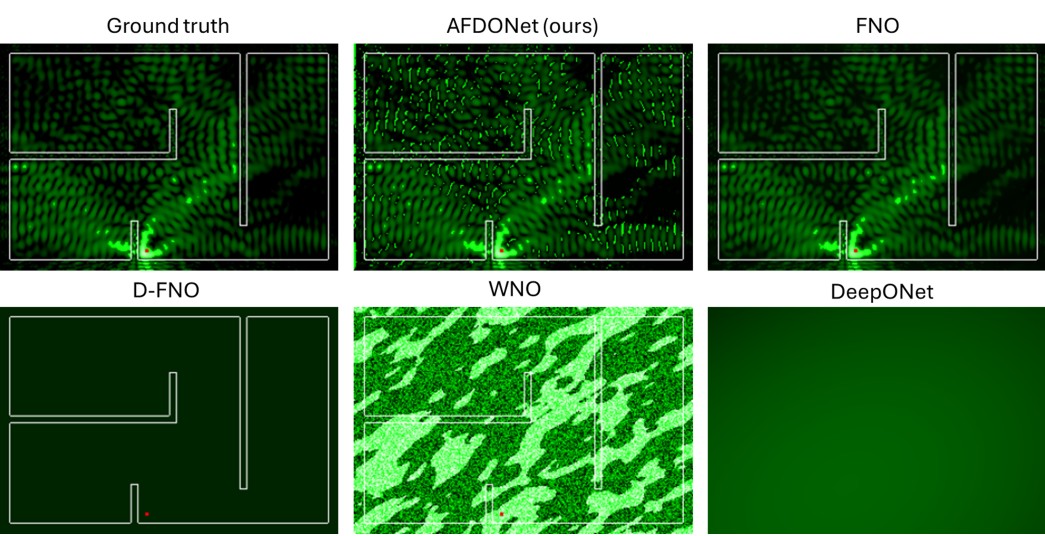

Figure 5: Ground truth and predicted solutions $u(x, y)$ of the Helmholtz equation on the planar manifold.

AFDONET PERFORMANCE ON NAVIER-STOKES EQUATION WITH RANDOMIZED VORTEX DATASET

We extend the ablation study shown in Table 2 with a new ablation study for the Navier-Stokes example with randomized vortex field dataset. The initial condition is set by vortex structures via Gaussian-based stream functions $\psi = A \cdot \exp\left(-\frac{(x-c_x)^2 + (y-c_y)^2}{2r^2}\right)$ with randomized parameters vortex centers $(c_x, c_y) \sim \mathcal{U}(1,5)^2$, radii $r \sim \mathcal{U}(0.5, 2)$, and strengths $A \sim \mathcal{U}(-2, 2)$.

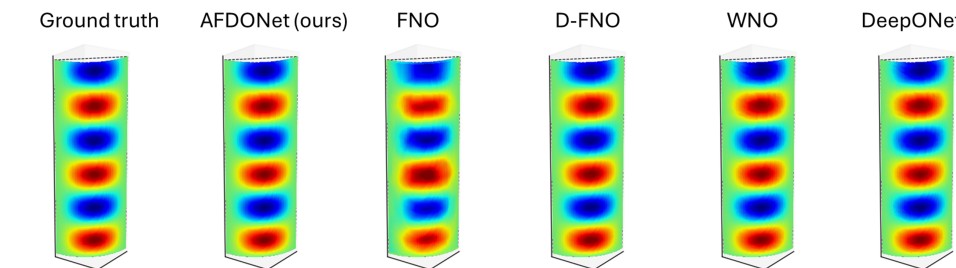

Figure 6: Ground truth and predicted solutions $u(\phi, z)$ of the Poisson equation on the quarter-cylindrical surface.

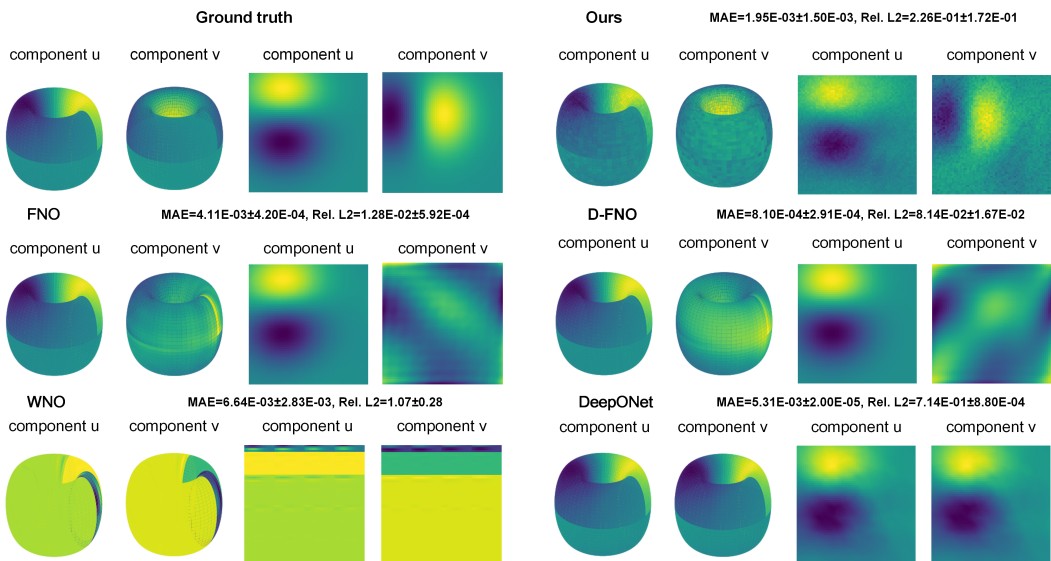

Figure 7: Ground truth and predicted fields $(u, v)$ of the Navier-Stokes equation (for static task) on both the torus $\mathbb{T}^2$ and the heatmap for various solvers. Here, the dataset is generated from Gaussian-based randomized vortex fields (dataset size is $5000$) (Pedergnana et al., 2020). Average MAE and relative $L^2$ errors and their standard deviations obtained using five random seeds are also reported.

# F ADDITIONAL EXPERIMENTS

## F.1 EXPERIMENT USING REAL-WORLD NOISY DATASET

To validate AFDONet's performance on noisy real-world datasets, we perform experiments using the latex glove DIC (Digital Image Correlation) original dataset (You et al., 2022). The goal is to learn the mechanical response of a nitrile glove sample directly from experimental data, without assuming a known constitutive law. The goal is to predict the displacement field at the current loading step. The input includes the spatial coordinates, the displacement field from the previous step, and the current boundary displacement. We compare the performance of AFDONet to the current SOTA of this dataset, IFNO, as well as FNO as follows. To ensure fair comparison, we conduct experiments using the same settings as IFNO with the number of hidden layers ranging from 3 to 12.

In addition, You et al. (2022) also reported the results of generalized Mooney-Rivlin (GMR) model in two settings. The relative $L^2$ errors of GMR model fitting and GMR inverse analysis are 3.30E-01 and 2.91E-01, respectively. We can observe that our AFDONet consistently outperforms other models in every $L$. Finally, the best reported result of IFNO is 3.30E-02 $\pm$ 4.63E-04 when $L = 24$ (You et al., 2022). Although we do not conduct the experiment $L = 24$ due to the limited time, our AFDONet still performs better than the best result of IFNO.

Table 6: Relative $L^2$ error of AFDONet and other baselines using the latex glove DIC (Digital Image Correlation) original dataset.

| Number of hidden layers | AFDONet | IFNO | FNO |
|---|---|---|---|
| 3 | 3.26E-02 $\pm$ 3.18E-04 | 3.43E-02 $\pm$ 4.96E-04 | 3.40E-02 $\pm$ 4.09E-04 |
| 6 | 2.78E-02 $\pm$ 4.01E-04 | 3.34E-02 $\pm$ 4.53E-04 | 3.84E-02 $\pm$ 4.21E-04 |
| 12 | 2.52E-02 $\pm$ 3.91E=04 | 3.32E-02 $\pm$ 4.41E-04 | 4.66E-02 $\pm$ 1.47E-03 |

The average training time of AFDONet is $\approx$5.3 seconds per epoch, which is comparable to that of IFNO ($\approx$4.6 seconds) and FNO ($\approx$5.7 seconds).

### F.2 PROBLEM DEFINED ON AN ARBITRARY MANIFOLD

To demonstrate the effectiveness of our AFDONet on arbitrary manifolds, here we design a new manifold that cannot be trivially projected onto a Euclidean space. The manifold is the closed unit ball $\overline{B} = \{z = (z_1, z_2) \in \mathbb{C}^2 : |z_1|^2 + |z_2|^2 \leq 1\}$, with boundary $\partial \overline{B} = S^3$. This is a compact 2-dimensional complex manifold equipped with the standard complex structure inherited from $\mathbb{C}^2$ and the flat Kähler metric $g = \sum_{j=1}^{2} dz_j \otimes d\bar{z}_j$. On this manifold, we solve the Schrödinger equation $(\Delta_A + q(|u|^2))u = 0$, where $\Delta_A = (d + iA)^*(d + iA)$ is the magnetic Laplacian with $d$ the exterior derivative, $*$ is the Hodge star with respect to the Kähler metric, $A$ is a smooth real-valued 1-form as the magnetic potential, and $q$ is a smooth complex-valued function as the electric potential. The results are shown below in Table F.2. Again, our AFDONet achieves significantly higher accuracy compared to baseline methods.

Table 7: Average MAE and Relative $L^2$ error of AFDONet and other baselines for solving the arbitrary manifold problem (values multiplied by 100).

| Metric | AFDONet (Ours) | FNO | D-FNO | WNO | DeepONet |
|---|---|---|---|---|---|
| MAE | $0.025 \pm 0.017$ | $4.506 \pm 0.927$ | $3.207 \pm 0.873$ | $5.884 \pm 1.374$ | $5.341 \pm 2.482$ |
| Rel. $L^2$ | $0.332 \pm 0.148$ | $55.688 \pm 5.415$ | $48.267 \pm 4.384$ | $99.99 \pm 0.000$ | $57.289 \pm 7.378$ |

