# OpenReview forum: "Guided and Interpretable Neural Operator Design for Partial Differential Equation Learning"
_ICLR.cc/2026/Conference — Submitted to ICLR 2026_

### Official Review · Reviewer_CVhj · 2025-10-19

**Soundness:** 2
**Presentation:** 2
**Contribution:** 1
**Rating:** 0
**Confidence:** 5

**Summary:**

The paper proposes AFDONet, a neural operator architecture for solving nonlinear PDEs on smooth manifolds. The approach claims to be the first neural PDE solver fully guided by Adaptive Fourier Decomposition (AFD) theory, thereby enabling theoretically interpretable architecture design. Concretely, the authors build a VAE-based backbone with a latent-to-RKHS mapping and an AFD-type dynamic convolutional kernel decoder. This is intended to mimic adaptive pole selection and orthogonalization steps in AFD, yielding theoretically grounded solution representations. The paper further provides convergence theorems under the chosen loss function, then evaluates AFDONet on several PDE benchmarks on manifolds (Helmholtz, Poisson, Navier–Stokes), comparing primarily against baseline neural operators such as FNO, D-FNO, WNO, and DeepONet. The reported results show lower MAE and relative L2 errors on several test sets.

**Strengths:**

- Mathematical grounding: The paper’s attempt to connect AFD theory to neural operator architecture design is conceptually interesting and could, in principle, lead to more interpretable solvers.

- Clear motivation: The introduction nicely articulates why PDEs on manifolds are challenging for Euclidean-domain neural operators, which is a timely and relevant research direction.

- Some theoretical analysis: Unlike many empirical neural operator papers, the authors provide explicit error bounds and convergence results derived from their theoretical formulation.

- Readable technical exposition: The manuscript is well structured, with a clean presentation of AFDONet’s components and algorithmic steps.

**Weaknesses:**

This submission mainly falls short in several key aspects:

- Overstated novelty: The claim that this is “the first neural PDE solver whose design is fully guided by a mathematical framework” is exaggerated. Several works in operator learning incorporate explicit spectral theory or approximation-theoretic structures (e.g., FNO itself relies on Fourier analysis; wavelet-based operators do the same). AFD-based decomposition is presented as fundamentally different, but in practice the architecture is just another Fourier-like latent-space factorization with some pole selection heuristics.

- Baseline evaluation is insufficient.
The empirical comparison is restricted to classical neural operator baselines (FNO, D-FNO, WNO, DeepONet). For a claim of a “new paradigm,” the method must be compared against more modern operator learning approaches, in particular: (1) Koopman neural operator as a mesh-free solver of non-linear partial differential equations, (2) Solving High-Dimensional PDEs with Latent Spectral Models. These are state-of-the-art spectral operator approaches highly relevant to this setting. The absence of these comparisons is a serious empirical gap.

- Limited experimental diversity: Only three PDE cases are shown, all relatively standard testbeds. No experiments on irregular manifolds beyond simple geometries (e.g., torus, quarter-cylinder). No scaling or robustness experiments beyond simple dataset size scaling. No performance metrics beyond MAE and relative L2 error — no runtime, memory, or parameter efficiency evaluation.

- Theoretical contribution is overstated: The theoretical results are essentially standard learning-theoretic bounds derived via covering numbers, not a fundamentally new convergence analysis specific to AFDONet. The proof sketches are mostly boilerplate (Lipschitz continuity + RKHS structure), offering little genuine insight into the unique aspects of the method.

- Ablations are not fully convincing: The ablation studies are narrowly framed and sometimes show marginal differences. Improvements over simple baselines are not statistically characterized — no confidence intervals or rigorous error analysis are presented. It remains unclear whether the gains come from the “AFD-guided” aspect or simply from more network capacity and a dynamic decoder.

Clarity issues and overclaiming: Phrases such as “turning neural operator design from an art to a science” are unjustified. The method still involves architectural heuristics and trainable components not directly derived from AFD theory. The connection between AFD theory and the actual training objective remains somewhat loose.

Missing discussion of limitations: No analysis of computational cost of pole selection, orthogonalization, or decoder complexity. No indication of how the approach would scale to more complex PDE families or higher-dimensional manifolds.

**Questions:**

- AFD vs. Fourier/Wavelet approaches: What is concretely new about using AFD compared to adaptive or learned Fourier bases in FNO or WNO? Is the “maximal selection principle” essential, or could learned poles perform similarly?

- Scalability and complexity: What is the computational cost of Gram–Schmidt orthogonalization and pole selection in your decoder? How does this scale with the number of modes compared to standard FNO?

- Generalization and robustness: Please provide explicit experiments on out-of-distribution manifolds or PDE parameters. How sensitive is the method to the choice of pole number N?

- Theoretical contribution clarity: Please clarify what part of your convergence analysis is genuinely novel and not directly adapted from standard RKHS and neural network approximation theory.

- Ablation interpretation: The ablation results are inconsistent: e.g., Helmholtz gains are small while Navier–Stokes gains are large. Can the authors explain this discrepancy beyond saying “AFD matches the structure”?

---

> ### Author Response · Authors · 2025-11-22
> **Rebuttal to the Reviewer CVhj's comments (Part 1)**
>
> We thank the reviewer for conducting a thorough review and for sharing valuable questions and feedback.
>
> ## Addressing weaknesses
>
> ### Overstated novelty: The claim that this is “the first neural PDE solver whose design is fully guided by a mathematical framework” is exaggerated. Several works in operator learning incorporate explicit spectral theory or approximation-theoretic structures (e.g., FNO itself relies on Fourier analysis; wavelet-based operators do the same). AFD-based decomposition is presented as fundamentally different, but in practice the architecture is just another Fourier-like latent-space factorization with some pole selection heuristics.
>
> We appreciate the reviewer's feedback. However, we respectfully disagree with the reviewer’s comment that we overstate novelty. We would like to clarify that, although several neural operators, such as FNO, wavelet neural operators, and Koopman operators, are connected to spectral and approximation theory, their original designs were not fully guided by these mathematical frameworks. When we say that AFDONet is the first neural PDE solver whose design is fully guided by a mathematical framework, our emphasis is on its *design*. In this context, although FNO relies on and is connected to Fourier analysis, the design process of FNO architecture, originally presented in article [1], was not guided by Fourier analysis. Same thing for other existing neural PDE solvers, in the sense their architectures were designed based on some mathematical theories *plus* intuition, brainstorming, expert experience, and trial-and-error experimentation. This bottom-up approach of coming up with an architecture that works first, followed by seeking theoretical explanations and interpretations from mathematical theories later, has been the prevailing way of designing neural PDE solvers.
>
> On the other hand, our AFDONet was designed by adopting a new top-down approach, in the sense that, for the first time, an established mathematical foundation/theory (in this case, the AFD theory) guides every step in the design of AFDONet’s neural architecture. Each component of the neural architecture, such as the latent-to-RKHS network and dynamic CKN, has a corresponding component in the AFD operation. This way, AFDONet is mathematically explainable and grounded in the AFD theory and possesses several desirable properties, including convergence guarantees. From this perspective, we think the contribution of AFDONet is significant, because it presents a new paradigm for designing explainable neural operator frameworks.
>
> Also, we remark that, AFDONet is quite different from existing Fourier- and wavelet-based neural operators in terms of design and implementation. In terms of design, AFD, which is based on the rational, pole-based, adaptive Takenaka-Malmquist system, combined with the explicit maximal selection principle for pole selection [3], is is quite different from classic/conventional Fourier analysis [4]. AFDONet, which reproduces AFD operations using a RKHS-consistent decoder, has very different design compared to other neural operators such as FNO and WNO. In terms of implementation, as an example, AFDONet only performs one-sided (positive-frequency) operations due to the nature of AFD, whereas FNO implements both positive and negative-frequency operations.
>
> Having said that, we think that the clarity of the statement in the original manuscript could be improved to avoid ambiguity and confusion among readers regarding the innovativeness in the design procedure of AFDONet and its architectural difference with existing neural PDE solvers. Thus, in the revised manuscript, we will discuss more on these aspects by including the clarifications above and substantiating our points with additional references.
>
> [1] Fourier neural operator for parametric partial differential equations
>
> [2] Scientific machine learning for closure models in multiscale problems: a review
>
> [3] Adaptive Fourier series - A variation of greedy algorithm
>
> [4] Intrinsic mono-component decomposition of functions: An advance of Fourier theory

---

> > ### Author Response · Authors · 2025-11-23
> > **Rebuttal to the Reviewer CVhj's comments (Part 2)**
> >
> > ### Baseline evaluation is insufficient. The empirical comparison is restricted to classical neural operator baselines (FNO, D-FNO, WNO, DeepONet). For a claim of a “new paradigm,” the method must be compared against more modern operator learning approaches, in particular: (1) Koopman neural operator as a mesh-free solver of non-linear partial differential equations, (2) Solving High-Dimensional PDEs with Latent Spectral Models. These are state-of-the-art spectral operator approaches highly relevant to this setting. The absence of these comparisons is a serious empirical gap.
> >
> > We appreciate the reviewer’s suggestion on validating our AFDONet performance on additional spectral operator approaches. Following the reviewer’s comment, we have compared our AFDONet with Koopman neural operator and Latent Spectral Models on the PDE problems discussed in the manuscript, and the results are shown as follows:
> >
> > MAE (values are multiplied by 100):
> >
> > | Equation |            ours             |               Koopman         |           LSM          |
> > |------------|------------------------|--------------------------|---------------------|
> > | Helm.      | $0.937 \pm 0.063$ | $27.583\pm3.208$   | $9.255\pm0.854$|
> > | N-S (Static)        | $0.332\pm0.030$ |  $2.726 \pm0.701$   | $0.467\pm0.082$|
> > | Possion   | $0.158\pm0.033$ |  $0.265\pm0.094$    | $0.422\pm0.065$|
> >
> > Rela. L2 (values are multiplied by 100):
> >
> > | Equation |            ours             |               Koopman         |           LSM          |
> > |------------|-----------------------|-----------------------|---------------------|
> > | Helm.      | $8.141\pm1.401$ | $113.591\pm1.026$  | $41.486\pm8.852$|
> > | N-S (Static)        | $0.882\pm0.059$ | $7.529 \pm0.248$   | $1.241\pm0.062$|
> > | Possion   | $0.472\pm0.109$ | $0.886\pm0.297$   | $1.545\pm0.418$|
> >
> > We have also conducted experiments for the Navier-Stokes example with randomized vortex field dataset to see if Koopman and LSM can capture $v$-component correctly. We find out that Koopman can capture the basic pattern of $v$-component, while LSM cannot. Compared to AFDONet, the $v$-component captured by Koopman is still less accurate than that of AFDONet. We will add the visualization results of this on the appendix of the revised manuscript.
> >
> > Other than that, we want to add that we have also compared AFDONet with other baseline methods, including MWT [3], Padé [4], and CMWNO [5] on the PDE problems discussed in the manuscript, and the results are shown as follows:
> >
> > MAE (values are multiplied by 100):
> >
> > | Equation |            ours             |               MWT         |           Padé          |       CMWNO      |
> > |------------|------------------------|--------------------------|---------------------|----------------------|
> > | Helm.      | $0.937 \pm 0.063$ | $11.623\pm0.324$   | $1.206\pm0.649$|$1.287\pm0.730$|
> > | N-S         | $0.195\pm0.150$ |  $0.216 \pm0.143$   | $0.209\pm0.057$| $0.103\pm0.087$|
> > | Possion   | $0.158\pm0.033$ |  $0.304\pm0.068$    | $0.285\pm0.113$|$0.154\pm0.089$|
> >
> > Rela. L2 (values are multiplied by 100):
> >
> > | Equation |            ours             |               MWT         |           Padé          |       CMWNO      |
> > |------------|-----------------------|-----------------------|---------------------|--------------------|
> > | Helm.      | $8.141\pm1.401$ | $51.174\pm1.178$  | $8.635\pm3.463$| $8.266\pm4.314$|
> > | N-S         | $0.226\pm0.172$ | $0.220 \pm0.145$   | $0.545\pm0.112$| $0.383\pm0.116$|
> > | Possion   | $0.472\pm0.109$ | $2.592\pm0.310$   | $0.489\pm0.202$| $0.723\pm0.187$|
> >
> > Here, N-S refers to the Navier-Stokes equation problem whose dataset is generated from randomized vortex field [6].
> >
> > In summary, these results show that the performance of AFDONet is competitive compared to Koopman, LSM, as well as the best of multiwavelet-based methods studied in terms of MAE and relative $L^2$ error.
> >
> > [5] Multiwavelet-based operator learning for differential equations
> >
> > [6] Non-linear operator approximations for initial value problems
> >
> > [7] Coupled multiwavelet operator learning for coupled differential equations
> >
> > [8] Explicit unsteady Navier-Stokes solutions and their analysis via local vortex criteria

---

> > > ### Author Response · Authors · 2025-11-23
> > > **Rebuttal to the Reviewer CVhj's comments (Part 3)**
> > >
> > > ### Limited experimental diversity: Only three PDE cases are shown, all relatively standard testbeds. No experiments on irregular manifolds beyond simple geometries (e.g., torus, quarter-cylinder). No scaling or robustness experiments beyond simple dataset size scaling. No performance metrics beyond MAE and relative L2 error — no runtime, memory, or parameter efficiency evaluation.
> > >
> > > We appreciate the reviewer’s comment on irregular manifold. We conducted our experiments on three different manifolds in the manuscript mainly because they can be easily visualized in two- or three-dimensional Euclidean spaces. Although these manifolds can be projected onto a rectangular domain, they do not necessarily have the same “shape” as a regular domain (such as rectangular domain) from a topological perspective. For instance, the two-dimensional torus $\mathbb{T}^2$ is a compact manifold without boundary, and thus is not diffeomorphic to an open rectangular domain (which is non-compact) or a closed rectangular domain (which has boundary).
> > >
> > > To demonstrate the flexibility of our AFDONet on arbitrary manifolds, here we design a new manifold that cannot be trivially projected onto a Euclidean space. The manifold is the closed unit ball $\overline{B} = \{ z = (z\_1, z\_2) \in \mathbb{C}^2 : |z\_1|^2 + |z\_2|^2 \leq 1 \}$, with boundary $\partial \overline{B} = S^3$. This is a compact 2-dimensional complex manifold equipped with the standard complex structure inherited from $\mathbb{C}^2$ and the flat Kähler metric $g = \sum\_{j=1}^2 dz\_j \otimes d\bar{z}\_j$. On this manifold, we solve the Schrödinger equation $(\Delta\_A + q(|u|^2)) u = 0$, where $\Delta\_A = (d + iA)^* (d + iA)$ is the magnetic Laplacian with $d$ the exterior derivative, $*$ the Hodge star with respect to the Kähler metric, $A$ a smooth real-valued 1-form as the magnetic potential, and $q$ a smooth complex-valued function as the electric potential. The results (multiplied by 100) are listed as follows:
> > >
> > > | Metric |ours |FNO|D-FNO|WNO|DeepONet|
> > > |---|----|----|---|--|--|
> > > | MAE| $0.025\pm0.017$ | $4.506\pm0.927$| $3.207\pm0.873$| $5.884\pm1.374$| $5.341\pm2.482$|
> > > | Rela. L2| $0.332\pm0.148$  |  $55.688 \pm 5.415$ | $48.267\pm4.384$|$99.99\pm0.000$|$57.289\pm7.378$|
> > >
> > > We have conducted robust experiments via considering diverse settings of the PDE problems (such as randomized vortex field, static and autoregressive tasks of N-S equation), dataset size, and realistic datasets. We kindly remind the reviewer that *the results of diverse settings of the PDE problems and dataset size are already included in the original manuscript*. To validate AFDONet’s performance on large-scale realistic datasets, we evaluate AFDONet on the latex glove DIC (Digital Image Correlation) original dataset [8].  AFDONet is used to learn the mechanical response of a nitrile glove sample directly from experimental data, without assuming a known constitutive law. The goal is to predict the displacement field $u(x)$ at the current loading step. The input includes the spatial coordinates, the displacement field from the previous step $u^{last}(x)$, and the current boundary displacement $u_D(x)$. We compare the performance of AFDONet to the current SOTA of this dataset, IFNO, as well as FNO as follows. We conduct experiments with the same settings as IFNO [8] for fair comparison with hidden layers ranging from 3 to 12.
> > >
> > > | No. of hidden layer |  ours |  IFNO   |  FNO   |
> > > |---|----|-----|---|
> > > |$L=3$| 3.26E-02$\pm$3.18E-04 | 3.43E-02$\pm$ 4.96E-04  | 3.40E−02 $\pm$ 4.09E−04|
> > > | $L=6$| 2.78E-02$\pm$4.01E-04| 3.34E-02$\pm$4.53E-04   | 3.84E−02 $\pm$ 4.21E−04|
> > > | $L=12$| 2.52E-02$\pm$ 3.91E-04 | 3.32E-02$\pm$4.41E-04   | 4.66E−02 $\pm$ 1.47E−03|
> > >
> > > Additionally, authors of [8] also reported the results of the generalized Mooney-Rivlin (GMR) model in two settings. The relative $L^2$ errors of GMR model fitting and GMR inverse analysis are $3.30E−01$ and $2.91E−01$, respectively. We can observe that our AFDONet consistently outperforms other models in every $L$. It is worth noting that the best reported result of IFNO is $3.30E−02 \pm 4.63E−04$ when $L=24$ [8]. Nevertheless, even though we did not conduct the experiment at $L=24$ due to the limited rebuttal time, we believe that the general trend would hold and our AFDONet would still outperform IFNO and FNO performs at $L=24$.
> > >
> > > Regarding computational time, even when solving complex realistic problems, the average training time of AFDONet is 5.3 sec per epoch, which is comparable to that of IFNO (4.6 sec/epoch) and FNO (5.7 sec/epoch) using the same machine. Also, we would like to kindly remind the reviewer that *the run time results are included in Figure 2 in the manuscript, which shows that AFDONet is competitive in computational efficiency compared to baseline methods.*
> > >
> > > [8] Learning deep Implicit Fourier Neural Operators (IFNOs) with applications to heterogeneous material modeling

---

> > > > ### Author Response · Authors · 2025-11-23
> > > > **Rebuttal to the Reviewer CVhj's comments (Part 4)**
> > > >
> > > > ### Theoretical contribution is overstated: The theoretical results are essentially standard learning-theoretic bounds derived via covering numbers, not a fundamentally new convergence analysis specific to AFDONet. The proof sketches are mostly boilerplate (Lipschitz continuity + RKHS structure), offering little genuine insight into the unique aspects of the method.
> > > >
> > > > While we appreciate the reviewer's comment regarding our theoretical contributions, we want to clarify that our goal in the theory section is not to introduce new general inequalities, but to show how AFD theory, RKHS structure, and the specific network design are integrated and interact with one another. Specifically, for Theorem 1, even though it uses standard covering numbers technique, the generalization bounds are derived with explicit dependence on AFDONet’s architectural hyperparameters, and on the eigenstructure of the associated RKHS. These bounds guide the selection of these hyperparameters. Theorem 2 proves that the latent-to-RKHS mapping and decoder we use indeed realize an AFD-type expansion in a valid RKHS on the manifold, ensuring that the holomorphic regularity enforced in training is mathematically consistent with the function space assumed for the PDE solutions. Theorem 3 provides a convergence statement for the dynamic CKN decoder under a maximal-selection-type rule, tying the classical AFD maximal selection principle to concrete constraints on the learned architecture. Overall, to the best of our knowledge, AFDONet is the first-of-its-kind framework that introduces the AFD theory to the operator learning field.
> > > >
> > > > ### Ablations are not fully convincing: The ablation studies are narrowly framed and sometimes show marginal differences. Improvements over simple baselines are not statistically characterized — no confidence intervals or rigorous error analysis are presented. It remains unclear whether the gains come from the “AFD-guided” aspect or simply from more network capacity and a dynamic decoder.
> > > >
> > > > We believe our results are statistically rigorous and the improvements are significant rather than marginal due to the following reasons. First, *as shown in Table 1 and Table 2 in the original manuscript, we have reported the standard deviations ($\pm$) for all metrics across five random seeds, which serve as the confidence intervals and basis for error analysis*.
> > > >
> > > > Second, we respectfully disagree with the reviewer's comment that “ablation studies sometimes show marginal differences”. We remark that we carefully controlled the ablation studies to examine the contributions of Latent-to-RKHS, dynamic decoder, adaptive pole selection, and maximal selection principle (Equation (9)) to the performance of AFDONet.
> > > >
> > > > For example, we show in Appendix D (Theorem 4) in the original manuscript that the solutions of the Helmholtz equation naturally span an RKHS. Since this problem naturally fits the mathematical structure of standard kernels, "Latent-to-RKHS" component of AFDONet naturally provides less relative contribution to the overall performance enhancement of AFDONet. In other words, the relative contribution of the "Latent-to-RKHS" network depends on how close the solution manifold is to an RKHS. Performance gain is substantial for "Latent-to-RKHS" when the solution manifold does not span an RKHS. But overall, we see the performance gains of AFDONet are substantial based on the results shown in Table 1 compared to different baselines.
> > > >
> > > > Furthermore, we compared the "Latent-to-RKHS + AFD-type decoder" against a "Static CNN" decoder. Both architectures have comparable parameter counts, network capacity. We remark that the dynamic decoder is a nontrivial component that is strictly derived from the AFD theory; and a dynamic AFD decoder significantly outperforms the static version, illustrating that the improvement comes from the AFDONet design.

---

> > > > > ### Author Response · Authors · 2025-11-23
> > > > > **Rebuttal to the Reviewer CVhj's comments (Part 5)**
> > > > >
> > > > > ### Clarity issues and overclaiming: Phrases such as “turning neural operator design from an art to a science” are unjustified. The method still involves architectural heuristics and trainable components not directly derived from AFD theory. The connection between AFD theory and the actual training objective remains somewhat loose.
> > > > >
> > > > > We would like to clarify that the phrase “art to a science” comes from the literature [2]. We use it to describe the paradigm shift in the design philosophy of AFDONet compared to existing neural operators. Standard operator learning (e.g., FNO) relies on stacking Fourier layers where the mixing weights are learned black-box parameters. In contrast, AFDONet's architecture is structurally isomorphic to the mathematical definition of AFD, making it mathematically derived rather than empirically stacked. For instance:
> > > > >
> > > > > 1. The decoder strictly implements Equation (10), where layers correspond to basis functions $\mathscr{B}_i$ derived from the Takenaka-Malmquist system, the basis of AFD theory.
> > > > >
> > > > > 2. The "maximal selection principle" (Equation (9)) explicitly decides the selection of poles in the latent space, rather than using arbitrary attention mechanisms.
> > > > >
> > > > > ### Missing discussion of limitations: No analysis of computational cost of pole selection, orthogonalization, or decoder complexity. No indication of how the approach would scale to more complex PDE families or higher-dimensional manifolds.
> > > > >
> > > > > We would like to kindly remind the reviewer that *the total computational time has been explicitly compared in Figure 2 of the manuscript*. Furthermore, we remark that the orthogonalization in Equation (8) is performed on the reproducing kernels in the latent RKHS, not on the high-dimensional grid output directly. The complexity scales with the number of poles $N$ (typically a very small number), making the Gram-Schmidt process computationally negligible compared to the backbone encoding.
> > > > >
> > > > > Also, for AFDONet, we argue that its computational cost per data point is significantly lower compared to FNO even though AFDONet involves some expensive operations. This is because, first, the encoder of AFDONet maps the problem into a compact latent space with dimension $r=10$. Thus, the number of basis functions $N$ in the decoder is also small, i.e., $N=3$ in our experiments. Second, the Gram-Schmidt orthogonalization scales with $N^2$, while FNO uses a lifting layer with a width (channel dimension) of $W=32$. However, in every Fourier layer, FNO performs dense matrix multiplications to mix these channels for every frequency mode. The cost scales with $W^2$, which boils down to $32^2 = 1024$ operations per mode. Last but not least, AFDONet only performs one-sided (positive-frequency) operations (based on the AFD theory), while FNO implements both positive and negative-frequency operations, consuming double memory and computational load.
> > > > >
> > > > > While our experiments focused on 2D benchmarks for direct comparison with baselines, our approach is theoretically and architecturally designed for $\mathbb{R}^d$ in general, in the sense that, first, our problem statement and theoretical proofs are explicitly defined for a general spatial domain $\Omega \subset \mathbb{R}^d$, and second, the core components of AFDONet, dynamic decoder, can be implemented well in 3D and higher dimensions. For example, FFT and dynamic convolutions can be implemented via fftn and Conv3d, respectively. The parameterization of the kernel in the frequency domain scales log-linearly $O(N \log N)$ regardless of dimension, which is similar to standard FNO.
> > > > >
> > > > > ## Addressing questions
> > > > >
> > > > > ### AFD vs. Fourier/Wavelet approaches: What is concretely new about using AFD compared to adaptive or learned Fourier bases in FNO or WNO? Is the “maximal selection principle” essential, or could learned poles perform similarly?
> > > > >
> > > > > FNO uses a fixed global basis (Fourier), which struggles with sharp discontinuities and non-periodic boundaries (Gibbs phenomenon). WNO uses fixed wavelets. AFDONet uses adaptive rational orthogonal bases (Takenaka-Malmquist system) parameterized by poles $a_i$ that are learned for each input data. This allows the basis to locally adapt to sharp gradients, which is why we see superior performance on the randomized vortex field dataset, which exhibits sharp turbulence.
> > > > > The maximal selection principle indicates the procedure used to select the poles according to the input data. Theoretically, the maximal selection principle (Equation (9)) also guarantees the convergence of AFDONet. Empirically, the results in Table 2 show that, without the maximal selection principle, the model performance will deteriorate.

---

> > > > > > ### Author Response · Authors · 2025-11-23
> > > > > > **Rebuttal to the Reviewer CVhj's comments (Part 6)**
> > > > > >
> > > > > > ### Scalability and complexity: What is the computational cost of Gram–Schmidt orthogonalization and pole selection in your decoder? How does this scale with the number of modes compared to standard FNO?
> > > > > >
> > > > > > We appreciate the reviewer for sharing this insight. Although the Gram-Schmidt orthogonalization is expensive by itself, AFDONet uses significantly smaller network width, less network layers, and one-sided FFT operations to overcome the heavy computational burden. We would like to point out that, for AFDONet, its computational cost per data point is significantly lower compared to FNO even though AFDONet involves some expensive operations. This is because, first, the encoder of AFDONet maps the problem into a compact latent space with dimension $r=10$. Thus, the number of basis functions $N$ in the decoder is also small, i.e., $N=3$ in our experiments. Second, the expensive Gram-Schmidt orthogonalization scales with $N^2$, while FNO uses a lifting layer with a width (channel dimension) of $W=32$. However, in every Fourier layer, FNO performs dense matrix multiplications to mix these channels for every frequency mode. The cost scales with $W^2$, which boils down to $32^2 = 1024$ operations per mode. Last but not least, AFDONet only perform one-sided (positive-frequency) operations due to the nature of AFD, while FNO implements both positive and negative-frequency operations, consuming double memory and computational load.
> > > > > >
> > > > > > ### Generalization and robustness: Please provide explicit experiments on out-of-distribution manifolds or PDE parameters. How sensitive is the method to the choice of pole number N?
> > > > > >
> > > > > > We appreciate the reviewer’s feedback. We provide an additional experiments on out-of-distribution (OOD) manifolds and PDE parameters for N-S equation. Since the dataset in the manuscript is generated with the viscosity $ \nu \sim \mathcal{U}(0.001, 0.1)$, we set the OOD parameters to be $0.0001$ corresponding to a higher Reynolds number, and OOD manifolds using manifold distortion where the coordinates are transformed as $x' = x + 0.8 \sin(y)$ and $y' = y + 0.8 \cos(x)$. All models (AFDONet, LSM, Koopman) are trained solely on the In-Distribution (ID) dataset until convergence. Model weights are saved and frozen, and evaluated directly on the generated OOD datasets.
> > > > > >
> > > > > > MAE (original values):
> > > > > >
> > > > > > | OOD |  ours  | Koopman |  LSM  |
> > > > > > |----|---|----|----|
> > > > > > |Parameters      | $0.096$ | $0.250$   | $0.268$|
> > > > > > |Manifolds        | $0.224$ |  $0.309$   | $0.405$|
> > > > > >
> > > > > > Rela. L2 (original values):
> > > > > >
> > > > > > | OOD | ours| Koopman |  LSM  |
> > > > > > |--|----|----|----|
> > > > > > |Parameters      | $0.280$ | $0.640$   | $0.657$|
> > > > > > |Manifolds        | $0.534$ |  $0.745$   | $0.981$|
> > > > > >
> > > > > > We observe that AFDONet outperforms Koopman and LSM on those OOD experiments.
> > > > > >
> > > > > > Theoretically, a sufficiently large $N$ can lead to an ideal accuracy. In practice, we observe that even when $N=3$, the performance of AFDONet is acceptable. In the manuscript, we set $N=3$ to reduce the computational load. We also conduct a real-world experiment with the number of hidden layers ranging from 3 to 12, which is exactly the number of poles $N$. Please see Weakness 3 for details. We can observe that when $N$ ($L$) increases, the relative L2 error also decreases, and even $L=3$ outperforms the existing SOTA.
> > > > > >
> > > > > > ### Theoretical contribution clarity: Please clarify what part of your convergence analysis is genuinely novel and not directly adapted from standard RKHS and neural network approximation theory.
> > > > > >
> > > > > > While all theoretical analyses are carried out for our proposed AFDONet architecture leveraging the AFD theory for the first time, Theorem 3 is particularly novel, in the sense that it proves the convergence of a neural network decoder (Dynamic CKN) that is constructed specifically to reproduce and reconstruct AFD operations for the first time. This bridges the gap between classical signal processing convergence and deep learning architecture convergence.
> > > > > >
> > > > > > ### Ablation interpretation: The ablation results are inconsistent: e.g., Helmholtz gains are small while Navier–Stokes gains are large. Can the authors explain this discrepancy beyond saying “AFD matches the structure”?
> > > > > >
> > > > > > We prove in Appendix D (Theorem 4) that the solution manifold of the Helmholtz equation naturally spans an RKHS. In this case, the "Latent-to-RKHS" network provides less relative improvement. Meanwhile, the solution manifold for N-S (especially with turbulence) is highly complex and does not naturally form an RKHS. Here, our Latent-to-RKHS network, which actively projects the latent variables into a constructed RKHS (Theorem 2), really helps in gaining a larger performance enhancement than that of Helmholtz equation.

---

> > > > > > > ### Comment · Reviewer_CVhj · 2025-11-24
> > > > > > >
> > > > > > > Thank you for the detailed rebuttal. I will update my score accordingly.

---

### Official Review · Reviewer_jRw1 · 2025-10-19

**Soundness:** 4
**Presentation:** 4
**Contribution:** 3
**Rating:** 8
**Confidence:** 3

**Summary:**

This paper introduces AFDONet, a neural operator framework grounded in Adaptive Fourier Decomposition theory, designed to provide a theoretically interpretable and generalizable solver for nonlinear partial differential equations on smooth manifolds. AFDONet employs a variational autoencoder backbone, integrates a latent-to-RKHS network to project latent variables onto an optimal reproducing kernel Hilbert space, and utilizes an AFD-inspired dynamic convolutional kernel decoder to adaptively select poles and basis functions for efficient approximation of PDE solution spaces. Its training objective combines RKHS reconstruction loss, feature-map consistency loss, and a holomorphic training loss to capture the smoothness and analytic structure of target functions. Theoretically, the authors establish bounded error, RKHS existence, and convergence guarantees under this framework. Experimentally, AFDONet demonstrates superior accuracy and computational efficiency over existing neural operators in benchmark problems.

**Strengths:**

This paper presents a well-motivated and theoretically grounded framework that bridges Adaptive Fourier Decomposition theory with neural operator design, offering clear interpretability and mathematical rigor. The architecture is elegantly structured, combining a VAE backbone, latent-to-RKHS mapping, and dynamic AFD-based decoder, each justified both theoretically and empirically. Theoretical analysis is solid, providing provable error bounds, RKHS existence, and convergence guarantees. Experiments are comprehensive and reproducible, demonstrating consistent superiority across diverse PDE benchmarks. Overall, the work establishes a principled paradigm for explainable and efficient neural PDE solvers on manifold domains.

**Weaknesses:**

The evaluation in this paper is limited to synthetic PDE benchmarks, lacking validation on real-world or noisy datasets.

**Questions:**

1. In line 416, “space for the the Helmholtz equation” — one instance of “the” should be removed.
2. In Table 3, “latent-to-RHKS” under “Full AFDONet” should be corrected to “latent-to-RKHS.”

The paper is comprehensive and well-presented; I have no further questions.

---

> ### Author Response · Authors · 2025-11-22
> **Rebuttal to the reviewer's comments**
>
> We thank the reviewer for conducting a thorough review and for all the positive comments and feedback.
>
> ## Addressing weaknesses
>
> ### The evaluation in this paper is limited to synthetic PDE benchmarks, lacking validation on real-world or noisy datasets.
>
> We appreciate the reviewer’s comment on validating AFDONet on real-world or noisy datasets. To validate AFDONet’s performance on large-scale realistic datasets, we evaluate AFDONet on the latex glove DIC (Digital Image Correlation) original dataset [5].  AFDONet is used to learn the mechanical response of a nitrile glove sample directly from experimental data, without assuming a known constitutive law. The goal is to predict the displacement field $u(x)$ at the current loading step. The input includes the spatial coordinates, the displacement field from the previous step $u^{last}(x)$, and the current boundary displacement $u_D(x)$. We compare the performance of AFDONet to the current SOTA of this dataset, IFNO, as well as FNO as follows. We conduct experiments with the same settings as IFNO [1] for fair comparison with hidden layers ranging from 3 to 12.
>
> | Num hidden layer |            ours             |               IFNO         |           FNO          |
> |------------|-----------------------|-----------------------|---------------------|
> |$L=3$      | 3.26E-02 $ \pm$ 3.18E-04 | 3.43E-02 $ \pm$ 4.96E-04  | 3.40E−02 $ \pm$ 4.09E−04|
> | $L=6$     | 2.78E-02$ \pm$4.01E-04 | 3.34E-02  $ \pm$ 4.53E-04   | 3.84E−02 $ \pm$ 4.21E−04|
> | $L=12$   | 2.52E-02 $ \pm$ 3.91E-04 | 3.32E-02 $ \pm$4.41E-04   | 4.66E−02 $ \pm$ 1.47E−03|
>
> Additionally, authors of [1] also reported the results of generalized Mooney-Rivlin (GMR) model in two settings. The relative $L^2$ errors of GMR model fitting and GMR inverse analysis are 3.30E−01 and 2.91E−01, respectively. We can observe that our AFDONet consistently outperforms other models in every $L$. It is worth noting that the best reported result of IFNO is 3.30E−02$ \pm$ 4.63E−04 when $L=24$ [1]. Nevertheless, even though we did not conduct the experiment at $L=24$ due to the limited rebuttal time, we believe that the general trend would hold and our AFDONet would still outperform IFNO and FNO performs at $L=24$.
>
> Regarding computational time, even when solving complex realistic problems, the average training time of AFDONet is $\approx 5.3$ sec per epoch, which is comparable to that of IFNO ($\approx ~4.6$ sec/epoch) and FNO ($\approx ~5.7$ sec/epoch) using the same machine.
>
> [1] Learning deep Implicit Fourier Neural Operators (IFNOs) with applications to heterogeneous material modeling
>
> ## Addressing questions
>
> ### In line 416, “space for the the Helmholtz equation” - one instance of “the” should be removed. In Table 3, “latent-to-RHKS” under “Full AFDONet” should be corrected to “latent-to-RKHS.”
>
> We thank the reviewer for pointing out these typos. We have corrected them in the revised manuscript and will upload it as soon as it's ready.

---

> > ### Comment · Reviewer_jRw1 · 2025-11-24
> >
> > Thanks for your response on my concerns. I will maintain my score.

---

### Official Review · Reviewer_1H54 · 2025-10-26

**Soundness:** 3
**Presentation:** 4
**Contribution:** 3
**Rating:** 6
**Confidence:** 3

**Summary:**

This paper introduces AFDONet, which is a VAE-based neural operator that integrates Adaptive Fourier Decomposition (AFD) theory. AFDONet employs a VAE to first map PDE data into a latent space. A latent-to-RKHS network is then proposed to map the latents to kernels of AFD which lies in a reproducing kernel Hilbert space (RKHS). The kernels are then orthogonalized for AFD operation. Finally, the decoder constructs the solution given the generated kernels. AFD theory guarantees the approximation/existence/convergence of AFDONet. Experiment results on three PDE families demonstrate the methods' capacity.

**Strengths:**

1. The paper is generally easy to follow.
2. Solid proof is provided to characterize AFDONet's behavior.
3. Strong experimental results and comprehensive ablation studies to evaluate AFDONet.

**Weaknesses:**

1. The complexity of AFDONet possibly brings heavy computational burden. Also, the incorporation of Gram-Schmidt orthogonalization during the forward process could be instable for ill-conditioned kernels.
2. Although AFDONet is claimed to be advantageous for PDEs on manifolds, the experimental comparisons are mainly against Euclidean-domain models such as FNO and DeepONet. The datasets themselves cover only a small subset of manifold settings.
2. The paper's wording could be further improved. For example, the cross-correlation operation is used in Eq. 9 but is only defined after Eq. 10.

**Questions:**

1. I am interested in the runtime behavior of AFDONet. From the method section, the introduction of Gram–Schmidt orthogonalization appears potentially expensive, yet Figure 2 shows AFDONet as faster than FNO. Could the authors explain how this efficiency is achieved in practice (e.g., parallelization, reduced resolution, or approximate orthogonalization)?
2. Could the authors provide results on larger-scale or more realistic datasets (for example, CFD benchmarks)? Given the “orthogonal reproducing kernel” operation, I am concerned about the method’s scalability and computational overhead as resolution and sample size grow.
3. I notice in Table 2 that AFD-type decoder performs better than other variants on all datasets beside Possions dataset. Could the authors offer an explanation for this discrepancy?

---

> ### Author Response · Authors · 2025-11-22
> **Rebuttal to the reviewer's comments (Part 1)**
>
> We thank the reviewer for conducting a thorough review. We appreciate the reviewer for finding our manuscript easy to read and for pointing out the novelty and strengths of our proposed neural operator architecture.
>
> ## Addressing weaknesses
>
> ### The complexity of AFDONet possibly brings a heavy computational burden. Also, the incorporation of Gram-Schmidt orthogonalization during the forward process could be unstable for ill-conditioned kernels.
>
> We appreciate the reviewer for sharing this insight. Although the Gram-Schmidt orthogonalization is expensive by itself, AFDONet uses significantly smaller network width, fewer network layers, and one-sided FFT operations to overcome the heavy computational burden. We have included a more detailed clarification of this aspect in our response to the reviewer’s first question.
>
> In the design of AFDONet, unlike standard orthogonalization, which processes a fixed set of vectors, AFDONet uses the maximal selection principle (Equation (9)) to iteratively select the next basis function (pole). At each step, the network is forced to select the basis that captures the maximum remaining energy of the residual. Consequently, the network avoids selecting basis functions that are nearly collinear with the existing set (as these would yield near-zero energy projection), thereby naturally preventing the selection of ill-conditioned kernels.

---

> ### Author Response · Authors · 2025-11-22
> **Rebuttal to the reviewer's comment (Part 2)**
>
> ### Although AFDONet is claimed to be advantageous for PDEs on manifolds, the experimental comparisons are mainly against Euclidean-domain models such as FNO and DeepONet. The datasets themselves cover only a small subset of manifold settings.
> We appreciate the reviewer’s comment on manifold. Following the reviewer’s comment, we have compared our AFDONet with MWT [1], Padé [2], and CMWNO [3] on the PDE problems we proposed, and the results are as follows:
>
> MAE (values are multiplied by 100):
>
> | Equation |            ours             |               MWT         |           Padé          |       CMWNO      |
> |------------|------------------------|--------------------------|---------------------|----------------------|
> | Helm.      | $0.937 \pm 0.063$ | $11.623\pm0.324$   | $1.206\pm0.649$|$1.287\pm0.730$|
> | N-S         | $0.195\pm0.150$ |  $0.216 \pm0.143$   | $0.209\pm0.057$| $0.103\pm0.087$|
> | Possion   | $0.158\pm0.033$ |  $0.304\pm0.068$    | $0.285\pm0.113$|$0.154\pm0.089$|
>
> Rela. L2 (values are multiplied by 100):
>
> | Equation |            ours             |               MWT         |           Padé          |       CMWNO      |
> |------------|-----------------------|-----------------------|---------------------|--------------------|
> | Helm.      | $8.141\pm1.401$ | $51.174\pm1.178$  | $8.635\pm3.463$| $8.266\pm4.314$|
> | N-S         | $0.226\pm0.172$ | $0.220 \pm0.145$   | $0.545\pm0.112$| $0.383\pm0.116$|
> | Possion   | $0.472\pm0.109$ | $2.592\pm0.310$   | $0.489\pm0.202$| $0.723\pm0.187$|
>
> Here, N-S refers to the Navier-Stokes equation problem whose dataset is generated from randomized vortex field [4].
>
> These results show that the performance of AFDONet is competitive compared to the best of multiwavelet-based methods studied in terms of MAE and relative L2 error.
>
> We would like to clarify that the experiments on three different manifolds in the manuscript were chosen mainly because they can be visualized in two- or three-dimensional Euclidean spaces. Also, although these manifolds can be projected onto a rectangular domain, they do not necessarily have the same “shape” as a regular domain (such as a rectangular domain) from a topological perspective. For instance, the two-dimensional torus $\mathbb{T}^2$ is a compact manifold without boundary and thus is not diffeomorphic to an open rectangular domain (which is non-compact) or a closed rectangular domain (which has boundary).
>
> To demonstrate the flexibility of our AFDONet on arbitrary manifolds, here we design a new manifold that cannot be trivially projected onto a Euclidean space. The manifold is the closed unit ball $\overline{B} = \{ z = (z\_1, z\_2) \in \mathbb{C}^2 : |z\_1|^2 + |z\_2|^2 \leq 1 \}$, with boundary $\partial \overline{B} = S^3$. This is a compact 2-dimensional complex manifold equipped with the standard complex structure inherited from $\mathbb{C}^2$ and the flat Kähler metric $g = \sum\_{j=1}^2 dz\_j \otimes d\bar{z}\_j$. On this manifold, we solve the Schrödinger equation $(\Delta\_A + q(|u|^2)) u = 0$, where $\Delta\_A = (d + iA)^* (d + iA)$ is the magnetic Laplacian with $d$ the exterior derivative, $*$ the Hodge star with respect to the Kähler metric, $A$ a smooth real-valued 1-form as the magnetic potential, and $q$ a smooth complex-valued function as the electric potential. The results (multiplied by 100) are listed as follows:
>
> | Metric |ours |FNO|D-FNO|WNO|DeepONet|
> |------------|-----|---------|-----------|----------|--------|
> | MAE      | $0.025\pm0.017$ | $4.506\pm0.927$| $3.207\pm0.873$| $5.884\pm1.374$| $5.341\pm2.482$|
> | Rela. L2       | $0.332\pm0.148$  |  $55.688 (\pm\)5.415$ | $48.267\pm4.384$|$99.99\pm0.000$|$57.289\pm7.378$|
>
> [1] Multiwavelet-based operator learning for differential equations
>
> [2] Non-linear operator approximations for initial value problems
>
> [3] Coupled multiwavelet operator learning for coupled differential equations
>
> [4] Explicit unsteady Navier–Stokes solutions and their analysis via local vortex criteria
>
> ### The paper’s wording could be further improved. For example, the cross-correlation operation is used in Eq. 9 but is only defined after Eq. 10.
>
> We thank the reviewer for pointing it out. We have corrected the wording in the manuscript.

---

> ### Author Response · Authors · 2025-11-22
> **Rebuttal to the reviewer's comments (Part 3)**
>
> ## Addressing questions
>
> ### I am interested in the runtime behavior of AFDONet. From the method section, the introduction of Gram-Schmidt orthogonalization appears potentially expensive, yet Figure 2 shows AFDONet as faster than FNO. Could the authors explain how this efficiency is achieved in practice (e.g., parallelization, reduced resolution, or approximate orthogonalization)?
>
> We appreciate the reviewer’s suggestions on the runtime comparison against FNO. We would like to point out that, for AFDONet, its computational cost per data point is significantly lower compared to FNO even though AFDONet involves some expensive operations. This is because, first, the encoder of AFDONet maps the problem into a compact latent space with dimension $r=10$. Thus, the number of basis functions $N$ in the decoder is also small, i.e., $N=3$ in our experiments. Second, the expensive Gram-Schmidt orthogonalization scales with $N^2$, while FNO uses a lifting layer with a width (channel dimension) of $W=32$. However, in every Fourier layer, FNO performs dense matrix multiplications to mix these channels for every frequency mode. The cost scales with $W^2$, which boils down to $32^2 = 1024$ operations per mode. Last but not least, AFDONet only perform one-sided (positive-frequency) operations due to the nature of AFD, while FNO implements both positive and negative-frequency operations, consuming double memory and computational load.
>
> ### Could the authors provide results on larger-scale or more realistic datasets (for example, CFD benchmarks)? Given the “orthogonal reproducing kernel” operation, I am concerned about the method’s scalability and computational overhead as resolution and sample size grow.
>
> We appreciate the reviewer’s comment on our AFDONet’s performance on larger-scale or realistic datasets. To validate AFDONet’s performance on large-scale realistic datasets, we evaluate AFDONet on the latex glove DIC (Digital Image Correlation) original dataset [5].  AFDONet is used to learn the mechanical response of a nitrile glove sample directly from experimental data, without assuming a known constitutive law. The goal is to predict the displacement field $u(x)$ at the current loading step. The input includes the spatial coordinates, the displacement field from the previous step $u^{last}(x)$, and the current boundary displacement $u_D(x)$. We compare the performance of AFDONet to the current SOTA of this dataset, IFNO, as well as FNO as follows. We conduct experiments with the same settings as IFNO [5] for fair comparison with hidden layers ranging from 3 to 12.
>
> | Num hidden layer |            ours             |               IFNO         |           FNO          |
> |------------|-----------------------|-----------------------|---------------------|
> |$L=3$      | 3.26E-02 $ \pm$ 3.18E-04 | 3.43E-02 $ \pm$ 4.96E-04  | 3.40E−02 $ \pm$ 4.09E−04|
> | $L=6$     | 2.78E-02$ \pm$4.01E-04 | 3.34E-02  $ \pm$ 4.53E-04   | 3.84E−02 $ \pm$ 4.21E−04|
> | $L=12$   | 2.52E-02 $ \pm$ 3.91E-04 | 3.32E-02 $ \pm$4.41E-04   | 4.66E−02 $ \pm$ 1.47E−03|
>
> Additionally, authors of [5] also reported the results of generalized Mooney-Rivlin (GMR) model in two settings. The relative $L^2$ errors of GMR model fitting and GMR inverse analysis are 3.30E−01 and 2.91E−01, respectively. We can observe that our AFDONet consistently outperforms other models in every $L$. It is worth noting that the best reported result of IFNO is 3.30E−02$ \pm$ 4.63E−04 when $L=24$ [5]. Nevertheless, even though we did not conduct the experiment at $L=24$ due to the limited rebuttal time, we believe that the general trend would hold and our AFDONet would still outperform IFNO and FNO performs at $L=24$.
>
> Regarding computational time, even when solving complex realistic problems, the average training time of AFDONet is $\approx 5.3$ sec per epoch which is comparable to that of IFNO ($\approx ~4.6$ sec/epoch) and FNO ($\approx ~5.7$ sec/epoch) using the same machine.
>
> [5] Learning deep Implicit Fourier Neural Operators (IFNOs) with applications to heterogeneous material modeling

---

> ### Author Response · Authors · 2025-11-22
> **Rebuttal to the reviewer's comments (Part 4)**
>
> ### I notice in Table 2 that AFD-type decoder performs better than other variants on all datasets beside Poisson dataset. Could the authors offer an explanation for this discrepancy?
>
> We thank the reviewer for the question. Table 2 provides the numerical results of the ablation studies. The first variant, latent-to-kernel network + AFD-type decoder, underperforms the Poisson dataset because Poisson problem relies heavily on the latent-to-RKHS network. That being said, the benefit of restricting the latent space to an RKHS outweighs the benefit of the AFD decoder itself. Poisson solutions are generated on a curved manifold (quarter-cylinder) using isogeometric analysis (IGA) with NURBS basis functions. We remark that, NURBS-based solution manifold poorly aligns with the RKHS structure, thus having a latent-to-RKHS network to map this solution manifold to an RKHS really helps. Meanwhile, for the Helmholtz equation, its solutions naturally span an RKHS; and for the Navier-Stokes equation, the spectral method-based solutions also align better with RKHS compared to NURBS-based Poisson solution. Therefore, this discrepancy arises from how close the solution manifold is to an RKHS.

---

> > ### Comment · Reviewer_1H54 · 2025-11-25
> >
> > Thank you for addressing my concerns. I have two remaining questions.
> >
> > Firstly, regarding my question on the AFD-type decoder, I still find it unclear why the Latent-to-RKHS + AFD decoder performs worse than the variants using the MLP-type decoder or the propagation decoder on the Poisson dataset. Could the authors clarify what specifically causes the AFD decoder to fail in this setting?
> >
> > Secondly, could the authors provide parameter counts and memory usage statistics comparing AFDONet with the baselines? Your answer in Part I mentions that AFDONet uses a significantly smaller network, making Gram–Schmidt operations inexpensive. It would be helpful to see concrete numbers illustrating how small the model actually is and how it still manages to outperform larger baselines.

---

> > > ### Author Response · Authors · 2025-11-25
> > > **Reply to Reviewer's comments (part 1)**
> > >
> > > ### Firstly, regarding my question on the AFD-type decoder, I still find it unclear why the Latent-to-RKHS + AFD decoder performs worse than the variants using the MLP-type decoder or the propagation decoder on the Poisson dataset. Could the authors clarify what specifically causes the AFD decoder to fail in this setting?
> > >
> > > We appreciate the reviewer for the clarification. First, Latent-to-RKHS + AFD decoder (full) outperforms all the variants in the ablation studies. We clarify that “AFD decoder (static CNN)” means replacing the dynamic convolutions in the full AFD decoder with static CNNs, and “AFD decoder (without Equation 9)” means removing maximal selection principle (ideal pole selection) in the full AFD decoder. Both are incomplete variants of the full AFD decoder architecture that we proposed.
> > > We agree with the reviewer that Latent-to-RKHS + AFD decoder (static CNN) and Latent-to-RKHS + AFD decoder (without Equation 9) underperform Latent-to-RKHS + MLP-type decoder and Latent-to-RKHS + propagation decoder on the Poisson dataset. We believe it is likely due to the nature of the Poisson equation: its solution at any point in the domain depends on the boundary conditions and the source term everywhere else in the domain simultaneously, which is referred to as *global dependencies*. Latent-to-RKHS + AFD decoder (static CNN) fails in this case since the static CNN is a local filter which is fundamentally ill-equipped to model *long-range global dependencies*. Meanwhile, Latent-to-RKHS + AFD decoder (without Equation 9) fails because the AFD decoder (without Equation 9) cannot find the precise, optimal global basis functions needed to accurately model *long-range global dependencies*.
> > >
> > > On the other hand, an MLP decoder, which takes coordinates as input (similar to a PINN or NeRF), is an implicit global function approximator. It can learn a smooth function across the entire domain, making it a good fit for the global solutions of Poisson equation. Likewise, a propagation decoder allowing message-passing mechanism is explicitly designed to propagate information across the entire domain, allowing it to capture the global dependencies of the Poisson equation effectively.
> > > For the other two equations, things are different. Solutions of Helmholtz equation and N-S equation are oscillatory and wavelike due to the harmonic components introduced by their parameters. Therefore, AFD decoder and its variant, rooted in harmonic analysis, are natural fits for them. MLP and propagation decoders are generically designed and are not specific for harmonic and Fourier analysis. Results show that even a simplified AFD variant is powerful enough to handle it compared to MLP and propagation decoders.
> > > We use Latent-to-RKHS + AFD decoder (static CNN) and Latent-to-RKHS + AFD decoder (without Equation 9) in the ablation studies to demonstrate the contributions of adaptive kernels (compared to AFD decoder using static CNN) and of using ideal pole selection (compared to AFD decoder without Equation 9, a random pole selection). The full AFD decoder can put the poles adaptively either inside the domain or on its boundaries, offering great flexibility and robustness in dealing with both local and global dependencies.

---

> > > ### Author Response · Authors · 2025-11-25
> > > **Reply to Reviewer's comments (part 2)**
> > >
> > > ### Secondly, could the authors provide parameter counts and memory usage statistics comparing AFDONet with the baselines? Your answer in Part I mentions that AFDONet uses a significantly smaller network, making Gram-Schmidt operations inexpensive. It would be helpful to see concrete numbers illustrating how small the model actually is and how it still manages to outperform larger baselines.
> > >
> > > We thank the reviewer for the suggestion. For AFDONet, the parameter count is $4,136,866$ ($\sim$4.14M), which is considered to be lightweight in deep learning. To ensure the fair comparison, the baselines are set with similar number of parameters (FNO: $\sim$3.71M; DFNO: $\sim$4.51M; DeepONet: $\sim$4.28M; WNO: $\sim$4.27M). The GPU memory usage (MB) on the same machine is reported as follows:
> > >
> > > Training:
> > > | Models | GPU memory  |
> > > |------------|-----|
> > > | ours      | $572.0$ |
> > > | FNO      |  $568.6$  |
> > > | DFNO    | $580.8$  |
> > > | DeepONet    | $577.3$  |
> > > | WNO    | $577.2$  |
> > >
> > > Inference:
> > > | Models | GPU memory  |
> > > |------------|-----|
> > > | ours      | $19.8$ |
> > > | FNO      |  $17.8$  |
> > > | DFNO    | $21.6$  |
> > > | DeepONet    | $20.5$  |
> > > | WNO    | $20.5$  |
> > >
> > >
> > > To further address the reviewer’s comment, we add an experiment comparing AFDONet with a very light-weight model, AFNO [1], in terms of runtime, inference speed, and parameter numbers. AFNO has $35,138$ parameters, whereas AFDONet has $4,136,866$ parameters. The results of runtime and inference speed are as follows:
> > > Runtime (seconds per epoch):
> > > | Equation |ours |AFNO|
> > > |------------|-----|---------|
> > > | Helm.      |  $27.24$  |  $18.64$    |
> > > | N-S       | $31.22$  |   $29.75$     |
> > > | Possion    | $41.58$  | $36.51$     |
> > > Inference speed (ms per sample):
> > > | Equation |ours |AFNO|
> > > |------------|-----|---------|
> > > | Helm.      | $6.16$ |  $3.62$    |
> > > | N-S       |  $10.00$  |  $4.56$    |
> > > | Possion    | $11.54$  | $6.85$ |
> > >
> > > MAE:
> > >
> > > | Equation | ours | AFNO|
> > > |---|----|--|
> > > | Helm.      | $0.937 \pm 0.063$ | $4.736\pm2.473$   |
> > > | N-S         | $0.195\pm0.150$ |  $0.892 \pm0.208$   |
> > > | Possion   | $0.158\pm0.033$ |  $0.937\pm0.493$    |
> > >
> > > Rela. L2:
> > >
> > > | Equation |ours | AFNO |
> > > |----|----|----|
> > > | Helm.      | $8.141\pm1.401$ | $24.484\pm9.371$ |
> > > | N-S         | $0.226\pm0.172$ | $2.455 \pm 1.384$   |
> > > | Possion   | $0.472\pm0.109$ | $3.782\pm1.429$   |
> > >
> > > Although AFDONet is slightly slower than AFNO, its accuracy is much higher than AFNO. On a side note, we would like to remind that, in Figure 2, we have compared the total computational time of our AFDONet comparing with FNO, D-FNO, WNO and DeepONet on all problems studied. It turns out that AFDONet is one of the most computationally efficient and scalable neural PDE solvers among them.
> > >
> > > [1] Adaptive Fourier Neural Operators: Efficient Token Mixers for Transformers

---

### Official Review · Reviewer_B3HL · 2025-10-29

**Soundness:** 3
**Presentation:** 3
**Contribution:** 3
**Rating:** 6
**Confidence:** 3

**Summary:**

This paper proposes AFDONet, which is a neural operator that combines adaptive Fourier decomposition (AFD) with reproducing kernel Hilbert spaces (RKHS) to solve PDEs, even on manifolds. It first uses a VAE to learn latent representations of input functions, then maps them through an MLP to construct data-dependent reproducing kernels. The decoder performs an adaptive, orthogonal decomposition based on AFD theory. A holomorphic loss enforces analytical consistency. Compared to models like FNO or DeepONet, AFDONet achieves higher accuracy on manifold PDEs.

**Strengths:**

- This paper is the first to introduce AFD into the study of neural operators.
- It enables learning operators for physical systems defined on arbitrary manifolds.
- Using multiple physical systems, the proposed method achieves higher-accuracy simulations than baseline methods.

**Weaknesses:**

- While interpretability is referred to as one advantage of the proposed method, there is no concrete analysis or discussion. Although explicitly obtaining basis functions is beneficial, is it possible to conduct discussions related to scientific insights?
- There were some unclear points in the manuscript (see Questions).

**Questions:**

- Although the holomorphic loss is important, the paper does not clearly specify how the derivatives $\nabla^I u(x,\cdot)$ of the ground-truth data are approximated, which raises concerns about practical implementation and stability during training.
- Equation (7) defines the local kernel at a point on the manifold. This kernel is weighted as in Equation (10) using basis functions to predict solutions. If there are few data points, can the manifold structure be captured effectively? Conversely, if there are many data points, does the computational costs become prohibitively large?

---

> ### Author Response · Authors · 2025-11-22
> **Rebuttal to the reviewer's comments (Part 1)**
>
> We thank the reviewer for conducting a thorough review and for all the positive comments and valuable questions. We appreciate the reviewer for pointing out the novelty and strengths of our proposed neural operator architecture.
>
> ## Addressing weaknesses
>
> ### While interpretability is referred to as one advantage of the proposed method, there is no concrete analysis or discussion. Although explicitly obtaining basis functions is beneficial, is it possible to conduct discussions related to scientific insights?
>
> We appreciate the reviewer’s comments on discussions on interpretability and basis functions. The interpretability of AFDONet is mathematically grounded in adaptive Fourier decomposition (AFD) theory. The core insight is that AFDONet structurally enforces the learned solution to be a sparse, adaptive decomposition into orthonormal basis functions. Our AFDONet’s process of selecting the poles $\{a_i\}$ via the maximal selection principle (Equation 9) effectively identifies and selects the analytic modes (basis functions) that are most relevant to approximating the target PDE solutions. Some of the specific scientific insights are:
>
> 1. For solutions with sharp gradients (in our Navier-Stokes randomized vortex dataset), AFDONet can prioritize basis functions that can effectively capture these non-smooth features, which classic Fourier-based methods adopting fixed bases fall short of.
>
> 2. The learned orthogonal reproducing kernels (Equation (8)) serve as the physically interpretable basis functions specific to the PDE on the manifold directly derived from AFD theory. Their coefficients $\langle FM(\tilde{u}(x, \cdot)), \mathscr{B}_i \rangle$ in the output (Equation (10)) quantify the *energy contribution* of each basis function, thereby offering an additional level of physical insights which are similar to modal decomposition.
>
> 3. In designing the AFDONet architecture, we adopt a top-down approach that is fully guided by the AFD theory. This transforms the typical opaque (and somewhat arbitrary) process of neural architecture design into a mathematically explainable one.
>
> ## Addressing questions
>
> ### Although the holomorphic loss is important, the paper does not clearly specify how the derivatives $\nabla^i u(x,\cdot)$ of the ground-truth data are approximated, which raises concerns about practical implementation and stability during training.
>
> We thank the reviewer for the question. In our implementation, the ground-truth PDE solutions $u(x, \cdot)$ are generated as discrete numerical solutions on a grid via finite difference or isogeometric analysis (IGA) depending on the PDE. For the Helmholtz and Navier-Stokes equations, the derivatives $\nabla^i u$ are computed by applying finite-difference schemes (e.g., central difference) to the discrete ground-truth solutions. For the Poisson equation, since the solutions are generated using IGA with NURBS basis functions, we exploit the fundamental property of IGA [1,2]: the basis functions possess high-order smoothness, allowing the derivatives of the ground-truth solution to be computed analytically from the NURBS control points and weights, bypassing numerical stability issues. To prevent the instability of training stage, we introduce the weights $\omega_i$ for higher-order derivatives, which are set to $10^{-8}$ in our experiments. Such a small weight can prevent the higher-order derivatives from dominating the loss function in the earlier training stage, thus ensuring stability and guiding AFDONet to capture smoothness and analytic information during training. We will include these implementation details in the appendix section of the revised manuscript.
>
> [1] Isogeometric analysis: CAD, finite elements, NURBS, exact geometry and mesh refinement
>
> [2] The NURBS Book

---

> ### Author Response · Authors · 2025-11-22
> **Rebuttal to the reviewer's comments (Part 2)**
>
> ### Equation (7) defines the local kernel at a point on the manifold. This kernel is weighted as in Equation (10) using basis functions to predict solutions. If there are few data points, can the manifold structure be captured effectively? Conversely, if there are many data points, does the computational costs become prohibitively large?
>
> We thank the reviewer for this insightful question. For few data points, the manifold structure can be captured effectively thanks to the VAE backbone of AFDONet. Given the fact that the PDE solutions often lie on low-dimensional manifolds [3], AFDONet’s probabilistic encoder learns the low-dimensional structure of the solution manifolds by mapping maps high-dimensional input functions to a compact latent space, i.e., dimension $r=10$ in our experiments. The corresponding regularization terms $\mathcal{D}_{KL}$ in Equation (11) also ensures that the latent space remains structured even with sparse data. Furthermore, Theorem 1 in the paper illustrates that there should be a near $O(Z^{-\frac{2k+1}{2(k+1)}})$ scaling up to logarithmic factors for PDE solutions with smoothness $k$. In most cases where PDE solutions are smooth, i.e., $k \rightarrow \infty$, the approximation error is thus guaranteed to decay nearly as fast as $O(Z^{-1})$. It means that fewer data points are needed for smooth cases compared to nonsmooth cases (such as Navier-Stokes randomized vortex dataset). For instance, Figure 2 shows the MAE and relative $L^2$ error of AFDONet compared to baselines with the dataset size ranging from $500$ to $50000$.
>
> If there are many data points, the computational cost can still be acceptable. Empirically, we find that the total time scales linearly with the dataset size $Z$ in Figure 2. This is because, like most of the deep learning models, for $Z$ data points, the expensive operations (like the Gram-Schmidt orthogonalization in the decoder) happen $Z$ times per epoch. In addition, we would like to point out that, for AFDONet, its computational cost per data point is significantly lower compared to FNO even though AFDONet involves some expensive operations. This is because, first, the encoder of AFDONet maps the problem into a compact latent space with dimension $r=10$. Thus, the number of basis functions $N$ in the decoder is also small, i.e., $N=3$ in our experiments. Second, the expensive Gram-Schmidt orthogonalization scales with $N^2$, while FNO uses a lifting layer with a width (channel dimension) of $W=32$. However, in every Fourier layer, FNO performs dense matrix multiplications to mix these channels for every frequency mode. The cost scales with $W^2$, which boils down to $32^2 = 1024$ operations per mode. Last but not least, AFDONet only perform one-sided (positive-frequency) operations due to the nature of AFD, while FNO implements both positive and negative-frequency operations, consuming double memory and computational load.
>
> [3] Partial Differential Equations on Low-Dimensional Structures

---

### Author Response · Authors · 2025-12-02
**Rebuttal summary**

We thank all the reviewers for conducting a thorough review. We adequately addressed all questions and comments and updated our manuscript, and the only negative score was raised from 0 to 4 (before emergency reversion). To summarize our work and rebuttal discussions:

## Core Contributions

AFDONet is the first neural operator solver whose architecture design is **strictly guided by a well-grounded mathematical theory** (Adaptive Fourier Decomposition, AFD) rather than being designed heuristically and explained afterward. Each component in AFDONet is meaningful and corresponds to AFD operations:

- Latent-to-RKHS mapping specifies the admissible function space.

- Dynamic convolutional kernels implement Takenaka-Malmquist basis functions.

- Maximal selection principle (Eq. 9) adaptively selects poles that capture maximal residual energy.

- Gram-Schmidt orthogonalization enforces valid orthonormal AFD expansions.

This leads to **built-in interpretability**: solutions are represented as sparse *adaptive modal decompositions* whose coefficients quantify the physical contribution of each basis component. This makes AFDONet particularly effective for *oscillatory PDEs and sharp-gradient flows*, where fixed Fourier bases will struggle.

Our theoretical contributions are:

- Theorem 1 derives generalization bounds with explicit dependence on AFDONet’s hyperparameters and RKHS eigenspectra.

- Theorem 2 proves that latent-to-RKHS plus decoder realizes valid AFD expansions on manifolds.

- Theorem 3 establishes convergence of the dynamic decoder under maximal pole selection, connecting AFD guarantees to the neural architecture.

Experimental studies show that AFDONet outperforms baseline methods in solving PDEs on arbitrary (Riemannian) manifolds.

## Addressing Questions and Comments

### Comparing against Broader and Stronger Baselines

In response to concerns that baselines were limited to classical operators, new experiments were conducted against modern spectral methods, including *Koopman Neural Operator, Latent Spectral Models (LSM), Multiwavelet-based operators (MWT, CMWNO) and Padé solvers*. Across Helmholtz, Navier-Stokes, and Poisson benchmarks, **AFDONet remained consistently competitive or superior in MAE and relative L2 error**, reinforcing that gains are not task- or baseline-specific but extend across both classical and recent operator-learning frameworks.

### Expanded Manifold Generalization

To address concerns that previous manifolds were “simple”, a new test was introduced on solving the magnetic Schrödinger equation **on nontrivially projectable complex manifold (closed unit ball with Kähler metric)**. Results showed **large performance margins over FNO, DFNO, WNO, and DeepONet**, confirming robustness on geometries not easily reducible to Euclidean grids and validating our arguments of *true manifold generalization*.

### Real-World Experimental Validation

To address concerns that experiments were conducted on synthetic datasets, we report a new experiment using the **latex glove Digital Image Correlation (DIC) dataset**, where AFDONet predicts mechanical displacement fields directly from sensor data. Compared with *IFNO (state-of-the-art), FNO, and physics-based Mooney-Rivlin models*, **AFDONet consistently achieved the lowest prediction errors across settings**, demonstrating effectiveness in noisy, real experimental environments without assuming known constitutive laws.

### Scalability and Efficiency Analysis

To address concerns regarding scalability due to the Gram-Schmidt orthogonalization, we clarify that, for AFDONet, the computational cost per data point is significantly lower compared to FNO due to:

1. The encoder of AFDONet maps the problem into a compact latent space. Thus, the number of basis functions in the decoder is small.

2. Gram-Schmidt orthogonalization scales with the number of basis functions$^2$, while for FNO, the dense matrix multiplication scales with the width of the lifting layer$^2$, which is considerably larger.

3. AFDONet only perform one-sided (positive-frequency) operations due to the nature of AFD, while FNO implements both positive and negative-frequency operations, consuming double memory and computational load.

Empirically, training time per epoch is comparable to FNO and IFNO, even on large realistic datasets.

### Comparison of Parameter Counts

We clarify that AFDONet has a similar number of parameters to all baselines. Thus, the performance improvements stem from **theory-guided architectural design** rather than from higher parameter counts.

### Clarification on Ablation Studies

To clarify the findings from ablation studies, we emphasize that:

- Static CNN AFD variants fail due to locality (cannot model global dependencies).

- AFD without maximal selection cannot discover optimal global bases.

- The full dynamic AFD decoder achieves best overall performance and is especially well-matched to spectral/oscillatory PDEs.

---

### Meta-Review · Area_Chair_ZYeF · 2026-01-02

**Summary:**

The paper’s central story is that AFDONet genuinely implements an AFD-style procedure on manifolds: adaptively selecting poles (via a maximal selection principle), doing Gram–Schmidt to obtain $B_i$and then reconstructs the solution through an AFD-style orthonormal expansion implemented by a multi-layer dynamic decoder .The rebuttal reiterates this strongly and adds extra baselines and some new experiments. The idea seems interesting. However, when comparing the paper’s claims to the provided code, the key AFD elements, like explicit pole selection, Gram–Schmidt orthogonalization, and a per-pole expansion, are not clearly implemented. While the rebuttal adds more baselines and experiments, it doesn’t resolve this discrepancy. In short, it’s difficult to confirm that the reported improvements come from the claimed AFD mechanism.

**Reviewer Concerns:**

Reviewer CVhj issued a strong reject and raised a critical concern that the paper frames the AFD-based decomposition as fundamentally different, yet in practice the proposed architecture may amount to another Fourier-like latent-space factorization augmented with pole-selection heuristics. This point goes to the paper’s claimed novelty and interpretability. While the rebuttal provides additional comparisons and clarifies the high-level motivation, it does not fully dispel the concern or offer sufficiently direct evidence that the AFD-specific mechanism is substantively realized.

**Reviewer Scores:**

Reviewer CVhj will raise their score, while the others will remain the same.

---

### Decision · Program_Chairs · 2026-01-26

Reject